# Purification of HCC-specific extracellular vesicles on nanosubstrates for early HCC detection by digital scoring

Na Sun [1,2], Yi-Te Lee [1], Ryan Y. Zhang [1], Rueihung Kao[1], Pai-Chi Teng[3], Yingying Yang [4], Peng Yang [1], Jasmine J. Wang[3], Matthew Smalley [1], Pin-Jung Chen [1], Minhyung Kim [5], Shih-Jie Chou[1], Lirong Bao[1], Jing Wang [1], Xinyue Zhang [1], Dongping Qi [1], Juvelyn Palomique[6], Nicolas Nissen[6], Steven-Huy B. Han[4], Saeed Sadeghi[4,7], Richard S. Finn[4,7], Sammy Saab[4], Ronald W. Busuttil[7,8], Daniela Markovic[9], David Elashoff[9], Hsiao-hua Yu [10], Huiying Li [1], Anthony P. Heaney[4], Edwin Posadas[3], Sungyong You [3,5], Ju Dong Yang [6,11], Renjun Pei [2✉], Vatche G. Agopian [7,8✉], Hsian-Rong Tseng [1,8✉] & Yazhen Zhu [1✉]

We report a covalent chemistry-based hepatocellular carcinoma (HCC)-specific extracellular vesicle (EV) purification system for early detection of HCC by performing digital scoring on the purified EVs. Earlier detection of HCC creates more opportunities for curative therapeutic interventions. EVs are present in circulation at relatively early stages of disease, providing potential opportunities for HCC early detection. We develop an HCC EV purification system (i.e., EV Click Chips) by synergistically integrating covalent chemistry-mediated EV capture/ release, multimarker antibody cocktails, nanostructured substrates, and microfluidic chaotic mixers. We then explore the translational potential of EV Click Chips using 158 plasma samples of HCC patients and control cohorts. The purified HCC EVs are subjected to reverse-transcription droplet digital PCR for quantification of 10 HCC-specific mRNA markers and computation of digital scoring. The HCC EV-derived molecular signatures exhibit great potential for noninvasive early detection of HCC from at-risk cirrhotic patients with an area under receiver operator characteristic curve of 0.93 (95% CI, 0.86 to 1.00; sensitivity = 94.4%, specificity = 88.5%).

[1] California NanoSystems Institute, Crump Institute for Molecular Imaging, Department of Molecular and Medical Pharmacology, University of California, Los Angeles, CA 90095, USA. [2] Key Laboratory for Nano-Bio Interface, Suzhou Institute of Nano-Tech and Nano-Bionics, University of Chinese Academy of Sciences, Chinese Academy of Sciences, 215123 Suzhou, P.R. China. [3] Samuel Oschin Comprehensive Cancer Institute, Cedars-Sinai Medical Center, Los Angeles, CA 90048, USA. [4] Department of Medicine, David Geffen School of Medicine, University of California, Los Angeles, CA 90095, USA. [5] Division of Cancer Biology and Therapeutics, Departments of Surgery, Cedars-Sinai Medical Center, Los Angeles, CA 90048, USA. [6] Comprehensive Transplant Center, Cedars-Sinai Medical Center, Los Angeles, CA 90048, USA. [7] Department of Surgery, David Geffen School of Medicine, University of California, Los Angeles, CA 90095, USA. [8] Jonsson Comprehensive Cancer Center, University of California, Los Angeles, CA 90095, USA. [9] Department of Medicine, Statistics Core, David Geffen School of Medicine, University of California, Los Angeles, CA 90095, USA. [10] Smart Organic Materials Laboratory, Institute of Chemistry, Academia Sinica, 11529 Taipei, Taiwan. [11] Division of Digestive and Liver Diseases, Cedars Sinai Medical Center, Los Angeles, CA 90048, USA. ✉email: rjpei2011@sinano.ac.cn; vagopian@mednet.ucla.edu; hrtseng@mednet.ucla.edu; yazhenzhu@mednet.ucla.edu

Hepatocellular carcinoma (HCC) is the fourth most common cause of cancer-related deaths worldwide[1]. The poor prognosis of HCC can be attributed to the fact that diagnosis is often made at a late stage in disease development[2,3]. Earlier detection of HCC is critical to reducing high HCC mortality rates, as potentially curative therapeutic interventions are available to treat early-stage HCC. Current American Association for the Study of Liver Disease (AASLD) guidelines[3] recommend biannual liver ultrasonography with or without serum alpha-fetoprotein (AFP) for at-risk patients with cirrhosis; however, ultrasound is not sensitive enough to detect early lesions, and the reported performance of AFP varies widely[4]. Thus, the development of noninvasive diagnostics for early-stage HCC may significantly benefit cirrhotic patients at risk for developing HCC.

Among the three conventional liquid biopsy[5,6] approaches in the context of oncology, i.e., circulating tumor cells (CTCs)[7–9], circulating tumor DNA (ctDNA)[10,11], and extracellular vesicles (EVs)[12], EVs are present in circulation at relatively early stages of disease[13] and persist across all disease stages. Furthermore, EVs' inherent stability guarantees the integrity of biomolecular cargos. Therefore, tumor-derived EVs can be regarded as "biomarker reservoirs"[14], promising downstream molecular analysis for noninvasive cancer diagnosis[15]. However, conventional EV isolation methods, e.g., ultracentrifugation[16], precipitation processing, and microfluidic enrichment[17–22] are based on EVs' physical properties (density, solubility, or size), and are incapable of separating tumor-derived EVs from total EVs. Since the majority of EVs in circulation are not of tumor origin, high background noise makes analysis of total EVs of limited diagnostic power[23]. To overcome this issue, our group[24] and others[23,25,26] have been exploring various immunoaffinity-based approaches to purify tumor-derived EVs. In parallel, others have been examining the potential of EVs and their mRNA cargos for HCC detection[27]. We envision a more sensitive and specific early HCC diagnostic assay can be achieved by (i) developing a rapid and effective HCC EV purification system using a multimarker cocktail to recognize, enrich, and recover HCC EVs secreted from highly heterogeneous HCC[28–30], and (ii) coupling downstream molecular profiling to obtain HCC EV-derived mRNA signatures capable of distinguishing early-stage HCC from at-risk cirrhotic patients.

In this study, we develop an HCC EV purification system (i.e., EV Click Chips, Fig. 1) by synergistically integrating four powerful approaches, including covalent chemistry-mediated EV capture/release, multimarker antibody cocktails[31], nanostructured substrates[32], and microfluidic chaotic mixers[33], paving the way for implementation of noninvasive detection of early-stage HCC. First, the covalent chemistry-mediated EV capture/release is built upon the combined use of click chemistry[34]-mediated EV capture and disulfide cleavage[35]-driven EV release in conjunction with an optimized multimarker cocktail targeting three HCC-associated surface markers[31], including EpCAM, ASGPR1, and CD147. Further, the incorporation of densely packed silicon nanowires substrates (SiNWS) dramatically increases the device surface area[32] contacting/interacting with EVs. Moreover, the microfluidic chaotic mixer made of polydimethylsiloxane (PDMS) facilitates repeated physical contact[36] between SiNWS and the flow-through HCC EVs, further enhancing the performance of EV capture. In contrast to previous antibody-mediated EV capture[24], a pair of highly reactive click chemistry motifs[37] i.e., tetrazine (Tz) and trans-cyclooctene (TCO), are grafted onto EV capture substrates (i.e., SiNWS, via surface modification) and HCC EVs (via TCO-capture agent conjugation), respectively. Subsequently, the click chemistry reaction between Tz-grafted SiNWS and TCO-grafted HCC EVs is rapid, specific, irreversible, and bioorthogonal[37], resulting in immobilization of the HCC EVs with improved capture efficiency and reduced background. After click chemistry-mediated HCC EV capture, exposure to a disulfide cleavage agent, 1,4-dithiothreitol (DTT)[38] leads to the prompt release of the HCC EVs from the SiNWS by breaking the embedded disulfide bond. Recognizing the dire need of practical methods to quantitatively assess the performance (EV recovery yield and recovery purity) of any given EV purification system, we pioneer a quantitative evaluation method for assessing the performance of EV Click Chip. By adopting this quantitative method throughout the optimization process, we are able to accurately determine the performance of EV Click Chips, achieving an optimal HCC EV purification condition that is later used in the preclinical study. Finally, to progress toward noninvasive HCC screening, we examine the potential of a streamlined HCC EV-based mRNA assay that couples EV Click Chips for purification of HCC EVs and reverse-transcription droplet digital PCR (RT-ddPCR) for quantification of 10 HCC-specific mRNA transcripts[39] using plasma samples from HCC patients and control cohorts. HCC EV-derived 10-gene molecular signatures exhibit great potential for noninvasive early detection of HCC from at-risk cirrhotic patients.

## Results

**The design and preparation of an EV click chip**. An EV Click Chip (Fig. 1) is composed of two functional components: (i) Tz-grafted SiNWS: a patterned SiNWS[32] covalently functionalized with disulfide bonds that link to terminal Tz motifs[40], and (ii) an overlaid PDMS chaotic mixer[41] (Supplementary Fig. 1), housed in a custom-designed microfluidic chip holder. The fabrication of Tz-grafted SiNWS began with introducing 10–15 μm densely packed Si nanowires (diameter = 100–200 nm) onto SiNWS, offering ~30 times more surface area (in contrast to a flat substrate) for facilitating click chemistry-mediated HCC EV capture. The incorporation of disulfide bonds and terminal Tz motifs onto SiNWS was carried out via a 3-step procedure[40] (Supplementary Fig. 2). To confirm successful preparation of Tz-grafted SiNWS, X-ray photoelectron spectroscopy (XPS) was employed to monitor functional group transformation at each step[9,40]. The passive mixing behavior of the flow-through EVs in EV Click Chips was simulated (Supplementary Fig. 3) via the combined use of computational fluid dynamics (CFD) and dissipative particle dynamics (DPD) models[24], offering a theoretical explanation on how the configuration of the EV Click Chip results in the enhanced physical contact[42] between TCO-grafted HCC EVs and Tz-grafted SiNWS.

**Preparation of artificial plasma samples**. To allow accurate evaluation of the performance of EV Click Chips throughout the optimization process, artificial plasma samples were prepared by spiking 10-μL aliquoted HepG2 cell-derived EVs (harvested by ultracentrifugation[43,44]) into 90-μL plasma from a female healthy donor. As shown in Fig. 2a, the presence of male HepG2 cell-line-derived EVs in female plasma allows exploitation of the sex-determining region Y (*SRY*) gene for reliable quantification of HepG2-derived HCC EVs in purified EV samples since the *SRY* gene is absent in female healthy donor's plasma.

**RT-ddPCR assay for quantification of EVs**. A RT-ddPCR assay in Fig. 2a was used to quantify the copy numbers of *SRY* and *C1orf101* transcripts (encoded on Chromosome Y and Chromosome 1, respectively) in the artificial plasma samples before and after purification by EV Click Chips. The results can be used to calculate the recovery yield and recovery purity throughout the optimization process. We denoted the copy numbers of *SRY* transcripts in the original 10-μL aliquoted HepG2 EVs and the EV Click Chip-recovered HepG2 EVs as *SRY* transcripts$_{ori-EV}$ and

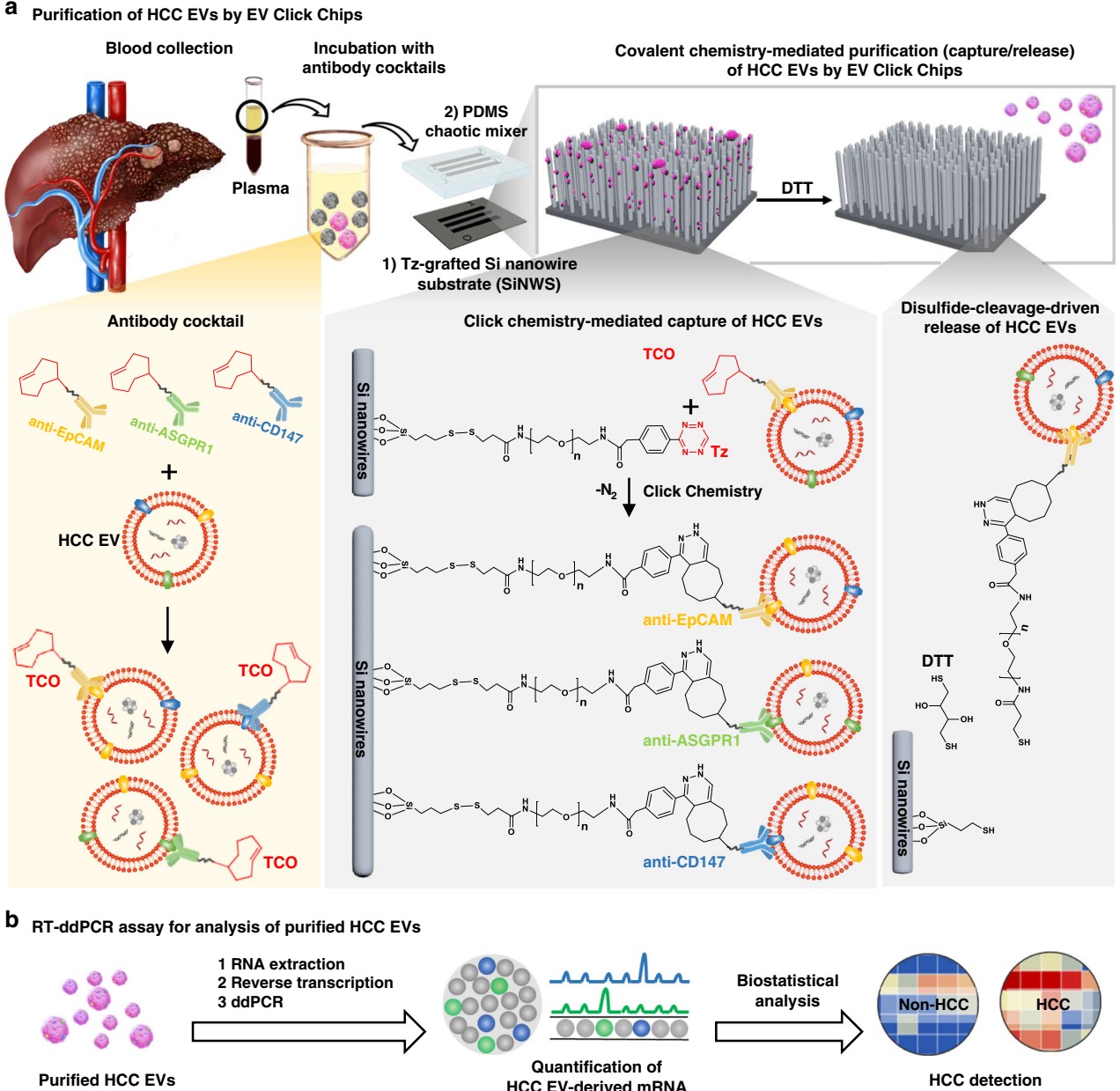

**Fig. 1 Purification and analysis of hepatocellular carcinoma extracellular vesicles (HCC EVs). a** Schematic illustration of the device configuration and working mechanism of an EV Click Chip, which is composed of a patterned Si nanowire substrate (SiNWS) covalently functionalized with tetrazine (Tz), and an overlaid polydimethylsiloxane (PDMS) chaotic mixer. The covalent chemistry-mediated EV purification approach combines the click chemistry-mediated EV capture and disulfide cleavage-driven EV release in conjunction with the use of an antibody cocktail targeting three HCC-associated surface markers, i.e., EpCAM, ASGPR1, and CD147. A pair of highly reactive click chemistry motifs, i.e., Tz and trans-cyclooctene (TCO), are grafted onto SiNWS and EVs, respectively. When a plasma sample flows through the device, click chemistry reaction between Tz-grafted SiNWS and TCO-grafted HCC EVs results in the immobilization of the HCC EVs. Subsequently, the exposure to 1,4-dithiothreitol (DTT) leads to the cleavage of the embedded disulfide bonds to release the immobilized HCC EVs. **b** The purified HCC EVs can then be subjected to reverse-transcription droplet digital PCR (RT-ddPCR) to obtain the signatures of 10 HCC-specific genes, which can be used to distinguish HCC patients from at-risk cirrhotic patients.

*SRY* transcripts$_{\text{rec-EV}}$, respectively. The EV recovery yield obtained by EV Click Chips under a given condition can be obtained from the following equation:

$$\text{HepG2 EV recovery yield} = \frac{SRY \text{ transcripts}_{\text{rec-EV}}}{SRY \text{ transcripts}_{\text{ori-EV}}} \quad (1)$$

In order to obtain the recovery purity of the EVs recovered by EV Click Chips, we first measured the intrinsic ratios between

*C1orf101* and *SRY* transcripts in aliquoted HepG2 EVs across a wide range of concentrations. As shown in Supplementary Fig. 4a, the ratios between *C1orf101* and *SRY* transcripts in HepG2 EVs exhibited a consistent linear correlation ($y = 1.95\ x$, $R^2 = 0.999$). With the *C1orf101*-to-*SRY* ratio determined as 1.95, we then calculated the recovery purity of the HepG2 EVs harvested from EV Click Chips as the ratio of the recovered *SRY* transcripts (contributed by recovered HepG2 EVs only) to the *C1orf101*

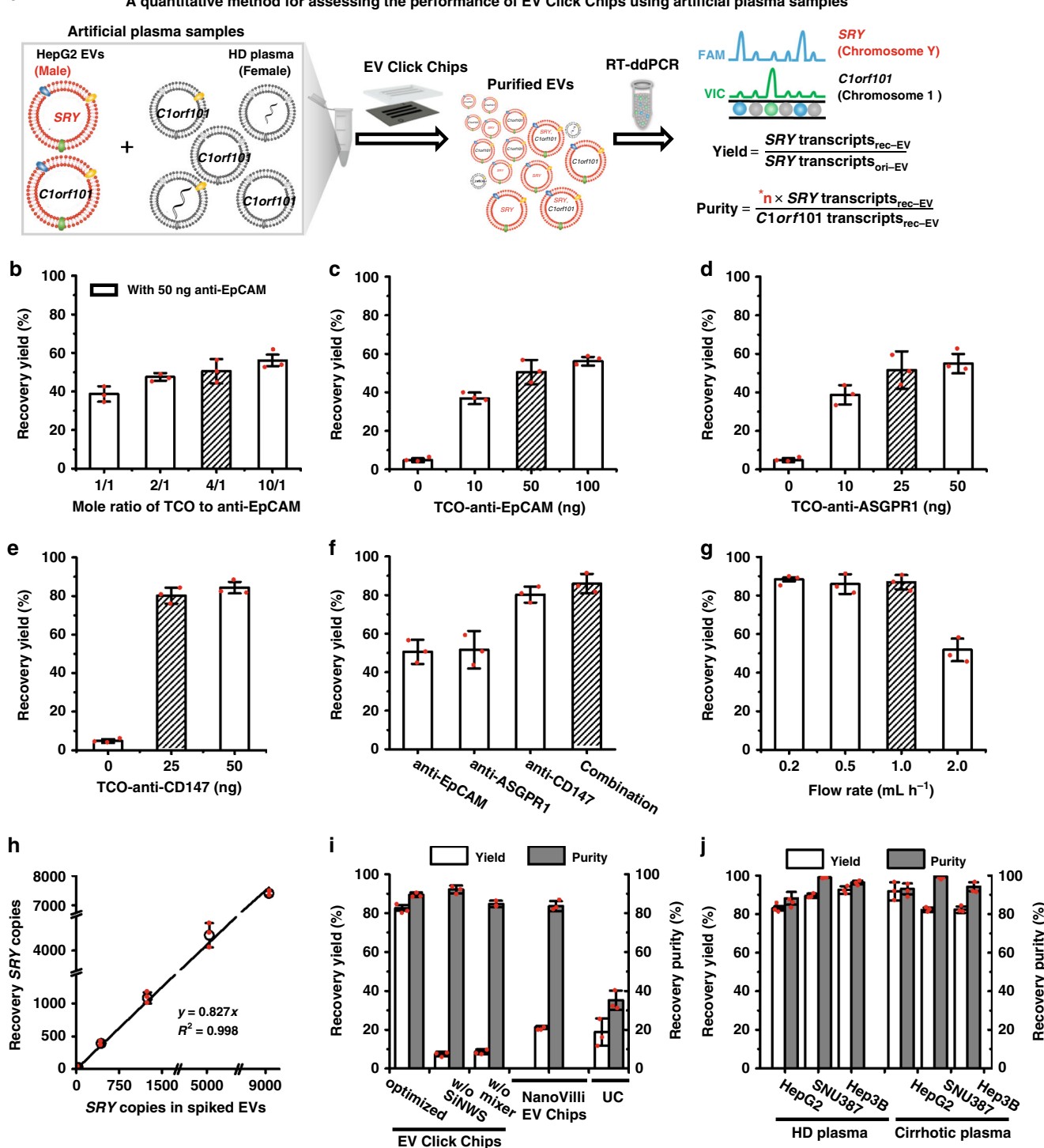

transcripts (contributed by both recovered HepG2 EVs and the nonspecifically captured background plasma-derived EVs, denoted as $C1orf101$ gene $_{rec-EV}$) using the following equation:

$$\text{HCC EV recover purity} = \frac{SRY \text{ transcripts}_{rec-EV}}{C1orf101 \text{ transcripts}_{rec-EV}} \times 1.95^* \quad (2)$$

$$^*1.95 \text{ is specific to HepG2 EVs}.$$

For HCC cell lines without $SRY$ transcripts, cancer-cell-derived EVs were spiked into plasma from male donors, and the EV recovery yield and recovery purity can be calculated using equations shown in Supplementary Methods and Supplementary Fig. 4b, c. A reproducibility study on the $C1orf101/SRY$ transcript

quantification methods used in the equations was conducted and the results are summarized in Supplementary Table 1.

**HCC EV purification with EV click chips**. Prior to conducting HCC EV purification (capture/release) studies, TCO motif was covalently conjugated onto each antibody agent (Fig. 1a), and the TCO-conjugated antibody agents were incubated with the artificial or clinical plasma samples for 30 min at room temperature. In each study (Fig. 2a), a 100-μL artificial plasma sample was introduced into an EV Click Chip, in which the click chemistry-mediated rapid and irreversible immobilization of HCC EVs on

**Fig. 2 Optimization of EV Click Chips using artificial plasma samples. a** A quantitative method was developed for evaluating the performance of EV Click Chips using artificial plasma samples prepared by spiking HepG2 EVs into the plasma from a female healthy donor (HD). A RT-ddPCR assay was employed to quantify the copy numbers of the *SRY* and *C1orf101* transcripts in the purified EV samples to calculate the recovery yield and recovery purity. *$n$ is the ratio between *C1orf101* and *SRY* transcripts in HepG2 EVs. **b** The recovery yields observed for EV Click Chips at different TCO-to-anti-EpCAM mole ratios. Data are presented as means ± SD of three independent assays. **c–f** The recovery yields obtained in the presence of individual and combined antibody capture agents, i.e., **c** anti-EpCAM, **d** anti-ASGPR1, **e** anti-CD147, and **f** combination of the three capture agents. Data are presented as means ± SD of three independent assays. **g** The recovery yields with different flow rates. Data are presented as means ± SD of three independent assays. **h** Dynamic ranges of EV recovery yields observed for EV Click Chips using artificial sample containing 0–9000 copies of *SRY* transcripts. Data are presented as means ± SD of three independent assays. **i** HepG2 EV recovery performance observed for (i) optimized EV Click Chips, devices without embedded silicon nanowires in SiNWS, and devices without herringbone features in the PDMS chaotic mixer, (ii) devices based on immunoaffinity EV capture (NanoVilli Chips) using the antibody cocktail concentration optimized for EV Click Chips, and (iii) ultracentrifugation (UC) approach. Data are presented as means ± SD of three independent assays. **j** General applicability of EV Click Chips for HCC EV recovery performance was validated using six artificial samples prepared by spiking three different HCC EVs (collected from HCC cell lines, i.e., HepG2, SNU387, and Hep3B) into two types of plasma samples (collected from either HD or liver cirrhotic patients). Data are presented as means ± SD of three independent assays.

SiNWS. Next, 100 μL DTT (50 mM) was introduced into the EV Click Chips to achieve disulfide cleavage-driven EV release. The DTT was removed in the subsequent RNA extraction process.

**A multimarker cocktail optimization for HCC EV capture.** Using published data from our group[31] and others[45,46], we identified surface markers that are highly expressed in HCC EVs, HCC CTCs, HCC cell lines, and primary tumor tissues of HCC patients, but virtually absent in white blood cells. Four candidate antibodies, i.e., anti-EpCAM, anti-ASGPR1, anti-CD147, and anti-GPC-3, against the corresponding surface markers were selected to achieve desired sensitivity and specificity for recognizing and capturing HCC EVs. The aforementioned RT-ddPCR assay was employed to assess the EV recovery yield of EV Click Chips in the presence of the individual antibodies and their cocktail mixtures. Figure 2b summarizes the recovery yields obtained by EV Click Chip at different TCO-to-anti-EpCAM mole ratios, and an optimal recovery yield was achieved at the TCO-to-anti-EpCAM ratio of 4:1. Under this TCO-to-antibody ratio, we suggest that the optimal amounts of individual candidate antibodies, i.e., anti-EpCAM (Fig. 2c), anti-ASGPR1 (Fig. 2d), anti-CD147 (Fig. 2e), and anti-GPC-3 (Supplementary Fig. 5a) are 50, 25, 25, and 50 ng, respectively. Using these optimized conditions, we compared the HCC EV recovery yields with different antibody cocktails. The data is summarized in Fig. 2f and Supplementary Fig. 5b and shows that the combination of anti-EpCAM, anti-CD147, and anti-ASGPR1 outperformed any single antibodies or other combinations.

**Optimization of EV Click Chips for HCC EV purification.** With the optimal antibody cocktail, flow rates of samples into EV Click Chips were studied, and >85% average recovery yields were observed at the flow rates of 0.2–1.0 mL h$^{-1}$ (Fig. 2g). To allow for a faster turnaround time for clinical samples, the flow rate of 1.0 mL h$^{-1}$ was selected. We then checked the dynamic range of EV Click Chips using artificial plasma samples spiked with different concentrations of EVs containing 0–9000 copies of *SRY* transcripts per 100-μL volume and confirmed the consistency of recovery yields ($y = 0.827x$, $R^2 = 0.998$) (Fig. 2h). To understand the crucial roles of the embedded silicon nanowires in SiNWS, the herringbone features in a PDMS chaotic mixer, and click chemistry-mediated EV capture, we carried out control experiments (Supplementary Fig. 6) using (i) the devices without embedded silicon nanowires in SiNWS or herringbone features in the PDMS chaotic mixer, and (ii) the devices based on immunoaffinity EV capture[24] (NanoVilli Chips), in parallel with EV Click Chips and the ultracentrifugation approach[44]. EV Click Chips exhibited a recovery yield of 82.7 ± 1.34% and recovery

purity of 90.2 ± 6.2%, which were significantly higher than those observed for the controls (Fig. 2i). The reproducibility of the EV Click Chips was evaluated by calculating the percent coefficient of variation (%CV) for recovery yields. The observed %CVs were calculated to be 1.12–12.65% for the intra-assay variability and 3.88 % for the inter-assay variability of the EV Click Chips (Supplementary Table 2). To test the general applicability of EV Click Chips and the optimized EV purification condition, the performance of EV Click Chips was further tested using six artificial samples prepared by spiking three different HCC EVs (collected from HCC cell lines, i.e., HepG2, SNU387, and Hep3B) into two types of plasma samples (collected from either healthy donors or liver cirrhotic patients). Detailed calculations of the reproducibility, recovery yields, and recovery purities for these artificial samples are described in Supplementary Table 3 and Supplementary Fig. 4. Overall, EV Click Chips achieved recovery yields ranging from 81.2 to 94.6% and purities ranging from 85.9 to 99.1% (Fig. 2j).

**Characterization of HCC EVs purified by EV click chips.** To better understand the working mechanisms of the click chemistry-mediated EV capture and disulfide cleavage-driven EV release, fluorescence microscopy, transmission electron microscopy (TEM), dynamic light scattering (DLS), and/or scanning electron microscopy (SEM) were employed to characterize the EV sizes and EV/SiNWS interfaces during the EV purification process, in which freshly harvested HepG2 EVs in PBS and healthy donors' plasma (Supplementary Fig. 7a–c) were used as a model system. To allow direct tracking of the capture and release processes of HCC EVs in EV Click Chips, HepG2 EVs were first labeled (Fig. 3a) with PKH26 dye (Sigma–Aldrich). The micrographs in Fig. 3b unveiled fluorescent signals on the SiNWS after EV capture and a dramatic signal reduction when the captured EVs were released by DTT. Figure 3c shows a representative TEM image of freshly harvested HepG2 EVs after uranyl acetate negative staining. These HepG2 EVs exhibited cup- or spherical-shaped morphologies with sizes ranging between 30 and 500 nm in diameter measured by TEM (inset of Fig. 3c). The size distributions of EVs measured by TEM were consistent with those observed by DLS (Supplementary Fig. 7d, e). Figure 3d shows a cross-sectional SEM image of Si nanowires with HepG2 EVs captured onto both the sidewalls (left) and the tops of the nanowires (right). After being released from EV Click Chips, the purified HepG2 EVs retained intact morphologies (Fig. 3e) with a similar size distribution (inset of Fig. 3e) to the freshly harvested HepG2 EVs. The purified HepG2 EVs from EV Click Chips were further verified by immunogold labeling with anti-CD63 (Supplementary Fig. 7f).

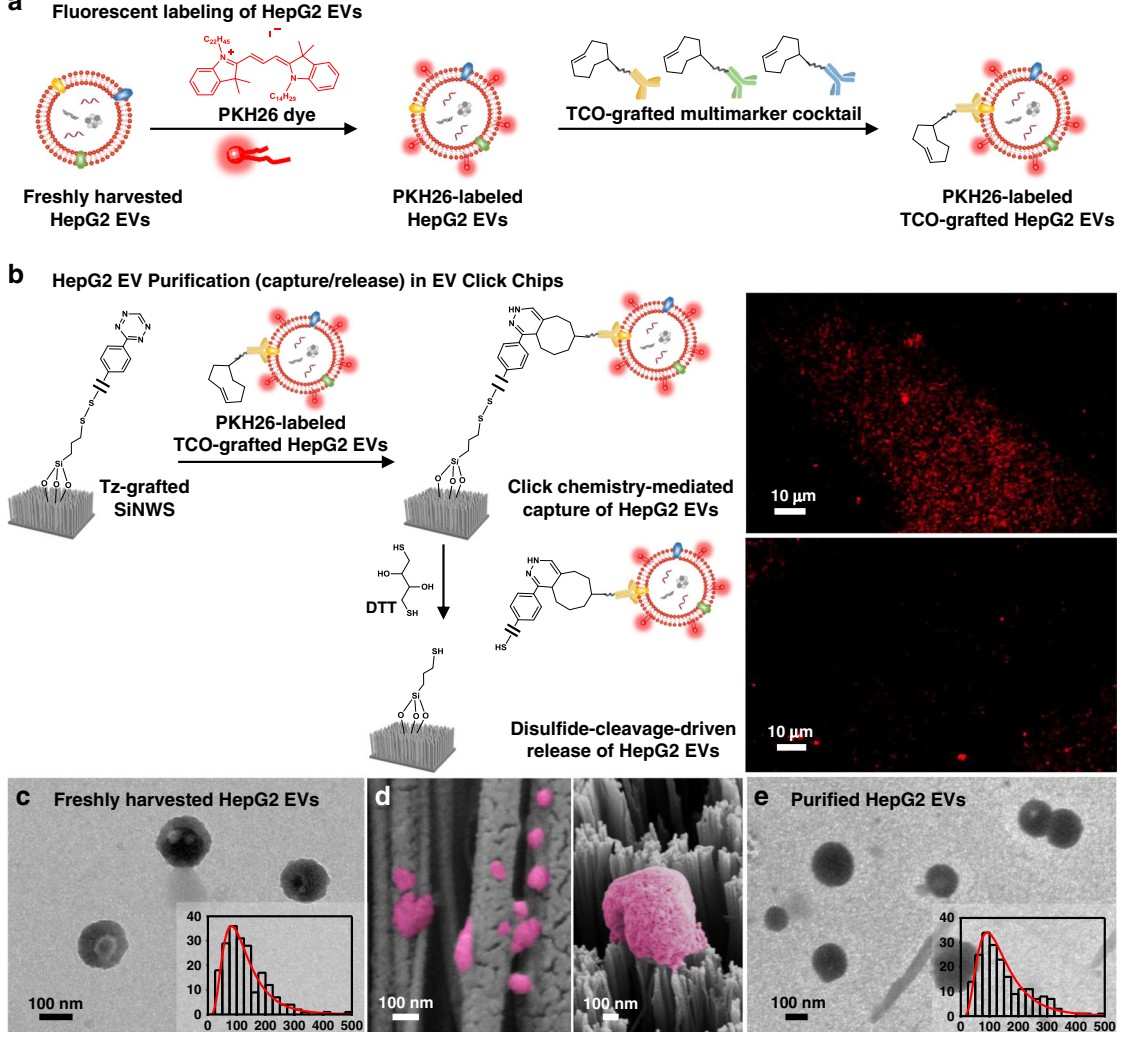

**Fig. 3 Characterization of HepG2 EVs purified by EV Click Chips. a** Fluorescent labeling of HepG2 EVs by PKH26 dye, followed by incubation with TCO-grafted antibody cocktail, giving PKH26-labeled TCO-grafted HepG2 EVs. **b** Tracking the purification (capture/release) process of HepG2 EVs in EV Click Chips using fluorescent microscopy. After click chemistry-mediated capture, PKH26-labeled HepG2 EVs were immobilized on SiNWS, as confirmed by the fluorescence micrograph (upper). Upon exposure to DTT, the surface linkers that anchored the PKH26-labeled HepG2 EVs onto SiNWS were cleaved, leading to the release of PKH26-labeled HepG2 EVs, as confirmed by fluorescence micrograph (lower). Data are representatives of three independent assays. **c**, Representative transmission electron microscopy (TEM) images of HepG2 EVs in bulk solution before capture. Inset: Size distribution ($n = 338$, diameters = 30–500 nm) of HepG2 EVs, measured by TEM. **d** Scanning electron microscopy (SEM) images of HepG2 EVs (colored in pink) on the sidewall (left) and tops (right) of the SiNWS. Data are representatives of three independent assays. **e** Representative TEM images of HepG2 EVs after being released from the chip. Inset: Size distribution ($n = 363$, diameters = 40–500 nm) of HepG2 EVs, measured by TEM.

**Quantification of 10 HCC-specific genes using purified HCC EVs.** By adopting the optimal HCC EV purification conditions, a workflow (Fig. 4a) for a streamlined HCC EV-based mRNA assay was developed by coupling EV Click Chips and RT-ddPCR for quantification of 10 well-validated HCC-specific mRNA transcripts[39] using clinical plasma samples. We collected 158 plasma samples from five cohorts, including (i) HCC cohort: newly diagnosed, treatment-naive HCC patients ($n = 46$, mean age = 66 y); (ii) cirrhosis cohort: patients with liver cirrhosis covering the etiology of hepatitis B virus (HBV), hepatitis C virus (HCV), alcoholic liver disease (ALD), and non-alcoholic steatohepatitis (NASH) ($n = 26$, mean age = 61 y). We confirmed that the cirrhosis cohort did not have HCC at the time of blood draw based on (1) negative multiphasic CT/MRI results, or (2) negative liver ultrasound results at the time of blood draw and a 6 month follow-up, or (3) observing no evidence of HCC on liver explant. (iii) hepatitis cohort: patients with chronic hepatitis B/C without

liver cirrhosis ($n = 25$, mean age = 57 y); (iv) healthy donors ($n = 23$, mean age = 52 y); (v) other cancer cohort: patients with primary malignancies other than HCC, with or without liver metastases ($n = 38$, mean age = 58 y). The clinical characteristics of these cohorts are provided in Supplementary Tables 4–8. Clinical annotation of all the plasma samples was performed by a clinician blinded to the assay. For each clinical sample, 0.5 mL of aliquoted plasma was introduced into an EV Click Chip to obtain purified HCC EVs. After RNA extraction, RNA concentrations were evaluated by Bioanalyzer 2100, (Supplementary Table 9), then RT-ddPCR was carried out to quantify the 10 HCC-specific genes, i.e., *alpha-fetoprotein (AFP)*, *glypican 3 (GPC3)*, *albumin (ALB)*, *apolipoprotein H (APOH)*, *fatty acid binding protein 1 (FABP1)*, *fibrinogen beta chain (FGB)*, *fibrinogen gamma chain (FGG)*, *alpha 2-HS glycoprotein (AHSG)*, *retinol binding protein 4 (RBP4)*, and *transferrin (TF)*[39]. We confirmed that these 10 mRNA markers are detectable in pure HepG2 EVs

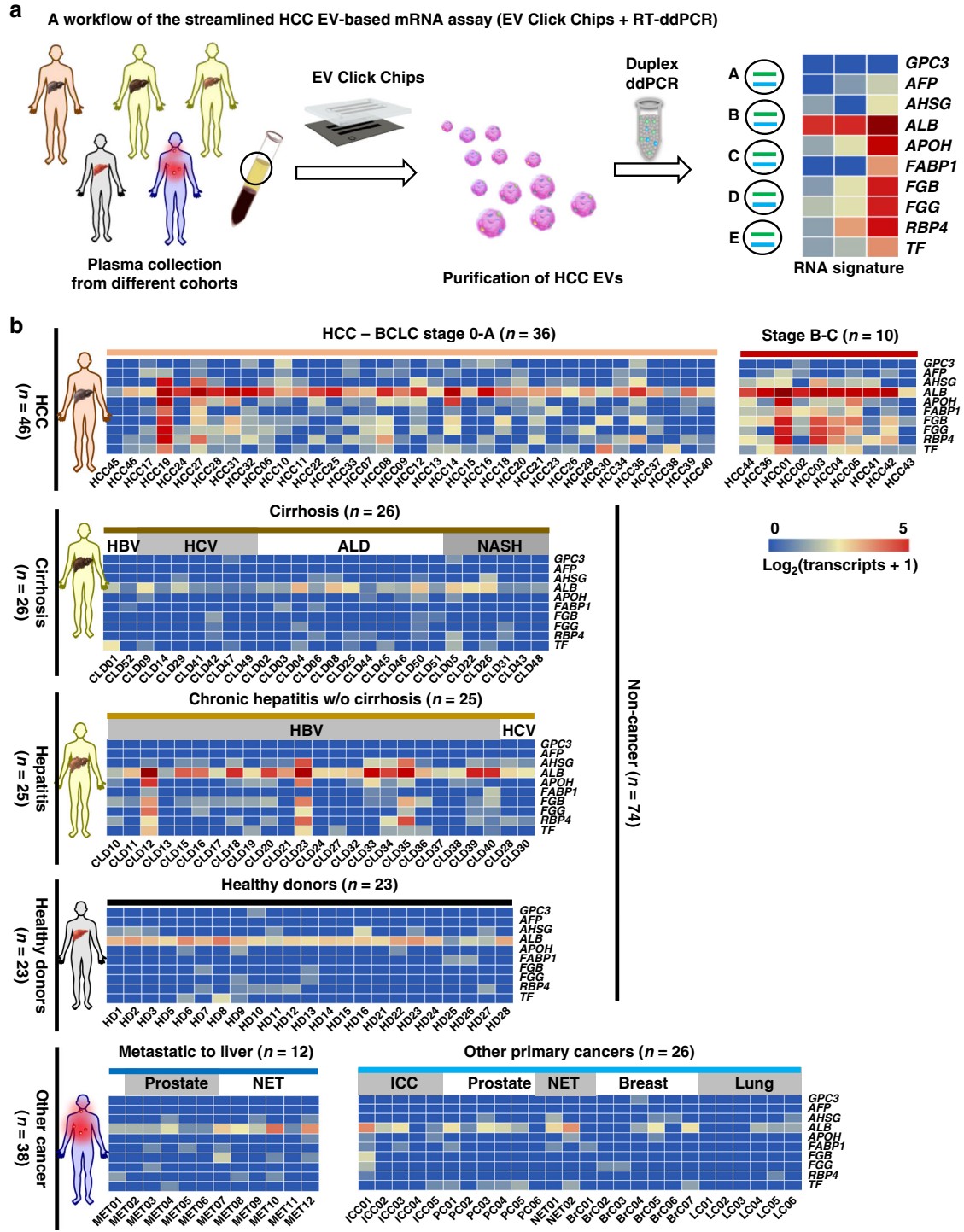

**Fig. 4 RT-ddPCR assay for quantification of 10 HCC-specific mRNA transcripts in purified HCC EVs. a** A general workflow developed for conducting HCC EV purification, followed by quantification of 10 HCC-specific mRNA transcripts in the purified HCC EVs. **b** Heatmaps depicting relative signal intensities for each gene expression of the 10 HCC-specific genes across different patient cohorts. (upper) Patients with newly diagnosed HCC (n = 46) are grouped according to Barcelona Clinic Liver Cancer (BCLC) staging system from early stages to advanced stages. (middle) Noncancer cohorts, including patients with liver cirrhosis (n = 26), chronic hepatitis (n = 25), and healthy donors (n = 23). (lower) Patients with cancers other than HCC (n = 38): cancers of nonhepatic origin metastatic to the liver (MET, n = 12); other primary cancers (n = 26), including intrahepatic cholangiocarcinoma (ICC), prostate cancer, midgut neuroendocrine tumor (NET), breast cancer, and lung cancer. Primary copy numbers are log2-transformed for each gene across all disease states. Clinical characteristics for each cohort are listed in Supplementary Tables 4–8. HBV hepatitis B virus, HCV hepatitis C virus, ALD alcoholic liver disease, NASH non-alcoholic steatohepatitis.

(Supplementary Fig. 8b). In addition, the publicly available EV databases, i.e. ExoCarta[47], Vesiclepedia[48], and exoRBase[49] also supported that these 10 mRNA markers are detectable in EVs. Considering EV-resident RNAs can be full-length or sometimes fragmented[50], the primers and probes of the 10 genes are specially designed to amplify short amplicons located 3'-most. To ensure the reproducibility of the ddPCR assay, we validated the PCR primers and probes using cDNA obtained from HepG2 cells, HepG2 EVs, and HCC EVs purified from five HCC patients' plasma samples by random priming reverse transcription (Supplementary Fig. 8). Finally, we summarized the HCC EV-derived 10-gene signatures obtained from 158 individual subjects in heatmaps (Fig. 4b); the primary copy numbers are log2-transformed for each gene across all disease states. As depicted in the heatmaps, higher signals were observed in the HCC cohort, compared with those from the noncancer cohorts (i.e., cirrhosis, hepatitis, and healthy donors) and other cancer cohort, including intrahepatic cholangiocarcinoma (ICC), breast cancer, lung cancer, prostate cancer, midgut neuroendocrine tumor (NET), and cancers of nonhepatic origin metastatic to the liver (MET). Furthermore, signal differences between early-stage and advanced-stage HCC patients defined by the Barcelona Clinic Liver Cancer (BCLC) staging system[51] allow for the separation of these two subgroups. Both Milan criteria[52] and United Network for Organ Sharing down-staging (UNOS DS)[53] criteria were also adopted to separate the HCC cohort into the respective early and advanced stages, and the results can be found in Supplementary Figure 9.

**HCC EV Z Scores for digital scoring.** We computed HCC EV Z Scores for each sample based on its 10-gene signatures in purified HCC EVs using the weighted Z-score method[54]. The copy numbers of the 10 genes were combined into the single HCC EV Z Scores. As depicted in the box plot (Fig. 5a), the HCC EV Z Score of the HCC cohorts (both early and advanced stages) are significantly higher (****$P < 0.0001$) than the noncancer cohorts (i.e., cirrhosis, hepatitis, and healthy donors) and other cancer cohort. HCC largely occurs in the setting of pre-existing chronic liver diseases[55]. However, it can also develop in the absence of such conditions. We thus performed receiver operator characteristic (ROC) analysis to test the potential of HCC EV Z Score for distinguishing HCC patients from noncancer patients (i.e., cirrhosis, hepatitis, and healthy donors). The area under the ROC curve (AUC) for distinguishing HCC from noncancer was 0.87 (95% CI, 0.80–0.94; sensitivity = 93.8%, specificity = 74.5%, Fig. 5b). Similarly, the potential of HCC EV Z Score for distinguishing HCC patients from primary malignancies other than HCC with or without liver metastases was then explored, and the AUC was 0.95 (95% CI, 0.90–1.00; sensitivity = 95.7%, specificity = 89.5%, Fig. 5c).

**HCC EV Z Scores for early HCC detection.** Finally, we examined the potential of HCC EV Z Score to distinguish early-stage HCC (BCLC stage 0-A) from at-risk liver cirrhosis. The AUC was 0.93 (95% CI, 0.86–1.00; sensitivity = 94.4%, specificity = 88.5%, Fig. 5d), which outperformed the most widely used serum

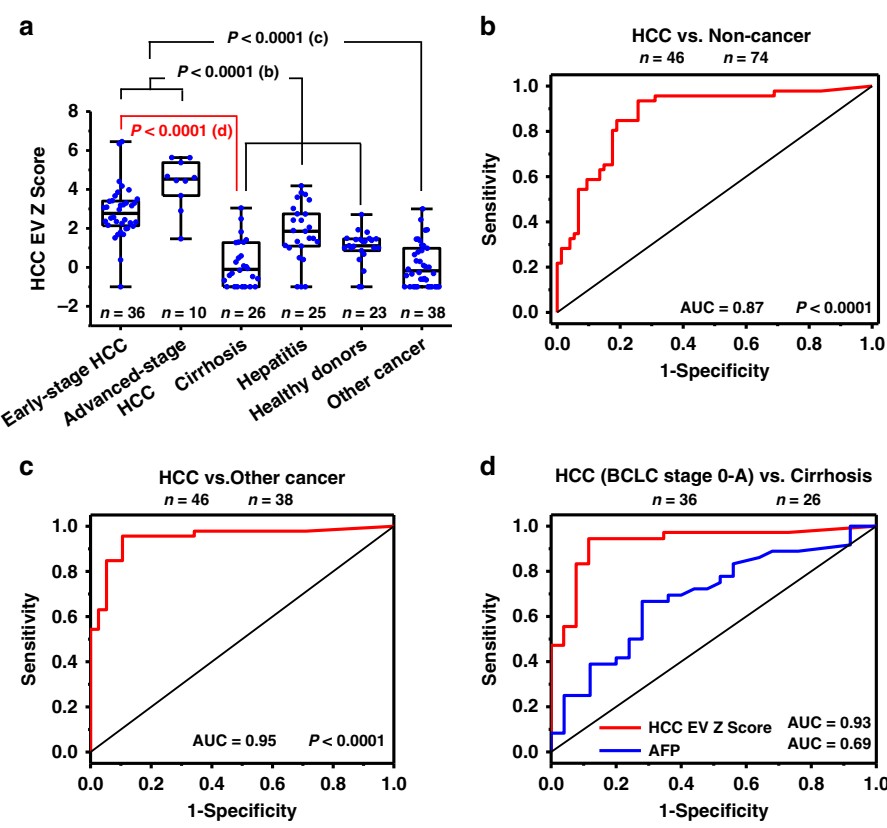

**Fig. 5 Statistical analysis on HCC EV Z Scores in different cohorts. a** Box plots representing the HCC EV Z Scores for different patient cohorts including early-stages HCC ($n = 36$), advanced-stage HCC ($n = 10$), cirrhosis ($n = 26$), hepatitis ($n = 25$), healthy donors ($n = 23$), and other cancers ($n = 38$). Whiskers ranging from minima to maxima, median and 25–75% IQR shown by box plots. Significant differences between different groups were evaluated using one-way ANOVA. **b, c** ROC curves for HCC EV Z Scores in **b** HCC versus noncancer (i.e., cirrhosis, hepatitis, and healthy donors) (AUC = 0.87, $P = 9.64E-12$, 95% CI, 0.80–0.94), **c** HCC versus other cancer (AUC = 0.95, $P = 1.79E-12$, 95% CI, 0.90–1.00). **d** ROC curves comparing HCC EV Z Scores (AUC = 0.93, $P = 1.02E-8$, 95% CI, 0.86–1.00) with the serum biomarker alpha-fetoprotein (AFP) level (AUC = 0.69, $P = 0.013$, 95% CI, 0.55–0.83) for differentiating early-stage HCC (BCLC, stage 0-A) vs. at-risk cirrhosis. Barcelona Clinic Liver Cancer (BCLC); ROC receiver operator characteristic.

biomarker AFP (AUCs of 0.69, 95% CI, 0.55–0.83; sensitivity = 66.7%, specificity = 72.0%) for early-stage HCC detection. The ROC curves in Supplementary Fig. 10 also demonstrated that HCC EV Z Score outperformed serum AFP testing in distinguishing early-stage HCC (defined by Milan[52] and UNOS DS[53] criteria) from at-risk liver cirrhosis with AUC of 0.91 versus 0.68, and 0.92 versus 0.70, respectively.

## Discussion

In this study, we have successfully developed and validated a HCC EV purification system, i.e., EV Click Chips, by uniquely integrating several coherent strategies including covalent chemistry-mediated EV capture/release, a multimarker antibody cocktail, nanostructured substrates, and a PDMS chaotic mixer, promising rapid and effective purification of HCC EVs with intact mRNA cargo. By coupling EV Click Chips with a downstream RT-ddPCR assay to quantify 10 well-validated HCC-specific mRNA transcripts[39], the resulting HCC EV-derived mRNA signatures exhibited great potential for noninvasive early detection of HCC.

The most unique feature of EV Click Chips is the exploration of the covalent chemistry-mediated EV purification through click chemistry-mediated EV capture and disulfide cleavage-driven EV release. We have demonstrated that the performance of click chemistry-mediated EV capture is superior to the immunoaffinity-based EV capture approaches on the same Nanostructured device, which are driven by the dynamic binding between a pair of antigens (on EVs) and antibodies (on the substrates). We attempted to address these issues by introducing click chemistry-mediated EV capture. Among different categories of click chemistry reactions, we selected the inverse-electron-demand Diels–Alder cycloaddition[56] between Tz and TCO motifs (a rate constant[57] of $10^4 M^{-1} s^{-1}$), considering their balanced chemical properties between reactivity and stability without the presence of a catalyst. The ligation between Tz-grafted SiNWS and TCO-grafted EVs is rapid, specific, irreversible, and insensitive to biomolecules, water, and oxygen, leading to immobilization of the EVs with improved capture efficiency and reduced nonspecific trapping of particles in the background.

Furthermore, because HCC EVs are secreted by highly heterogeneous HCC[28–30] cells, it is likely that a single capture agent would not provide sufficient performance for capture of HCC EVs. Therefore, it is necessary to develop an antibody cocktail to recognize and capture HCC EVs from clinical samples, allowing for sensitive and specific detection of HCC-derived EVs across all disease stages. Our experimental data using both artificial and clinical plasma samples showed that significantly greater EV capture yields and purities were achieved when utilizing a 3-antibody combination cocktail compared to every single antibody alone (i.e., anti-EpCAM, anti-ASGPR1, and anti-CD147).

Moreover, based on our previous experience in exploring the combined use of nanostructured immunoaffinity substrates and PDMS chaotic mixers to achieve highly efficient capture of targeted particles (i.e., CTCs and EVs) in peripheral blood, integrating this device configuration with click chemistry-mediated EV capture and a multimarker antibody cocktail offers a sensitive and specific technology for capturing HCC EVs with minimal background. This integration also permits effective conjugation of the TCO-grafted antibody cocktail onto the majority of HCC EVs in a small volume of solution, facilitating the click chemistry-mediated HCC EV capture onto EV Click Chips. Following HCC EV capture, subsequent disulfide cleavage-driven HCC EV release confers a second layer of specificity to the HCC EV purification process and improves the recovery purity of HCC EVs.

The combined use of a multimarker antibody cocktail and EV Click Chips could possibly lead to recovering EVs which are not of HCC origin. For example, anti-EpCAM could capture EVs from other epithelial tissues. To address this concern, we adopted the RT-ddPCR assay, which is capable of quantifying HCC-specific genes as a downstream readout for the purified HCC EVs. These 10 HCC-specific genes were selected from tissue lineage-associated transcripts expressed in liver cells but absent in blood cells and other tissues. Therefore, the resulting 10-gene signatures were predominantly contributed by HCC EVs, conferring a third layer of specificity to the streamlined HCC EV-based mRNA assay.

In the process of optimizing the EV Click Chip, we developed a simple and versatile quantitative evaluation method that has addressed the dire need of assessing the purification performance (EV recovery yield and recovery purity) of the EV Click Chip. Due to the lack of highly prevalent mutations in HCC, we devised a method where the SRY gene encoded on Chromosome Y from a male HCC cell line would be utilized as a surrogate HCC marker. An artificial plasma sample was prepared by spiking EVs from a male HCC cell line (e.g., HepG2) into plasma from a female healthy donor, and RT-ddPCR was adopted to count the copy numbers of the target SRY and the reference C1orf101 transcripts (encoded on Chromosome Y and Chromosome 1, respectively) for distinguishing and quantifying the spiked HCC EVs. This method is more convenient and quantitative than existing methods[23] that required prelabeling or pretransfection of EVs with specific transcripts. This method is also broadly applicable to the optimization of any other tumor-derived EV purification platform prior to clinical study.

There have been promises on the horizon for emerging liquid biopsy-based HCC diagnostics such as ctDNA-based methylation for HCC detection[58] and CTC-based RNA signature for HCC detection[39]. Although ctDNA methylation profiling using whole genome bisulfite sequencing can detect early-stage HCC[58], its use in HCC screening may be challenging because of the relatively high cost and long turnaround time. On the other hand, CTCs seem to enable high specificity detection of HCC-specific mRNA signatures, but current data[39] has shown that the sensitivity of CTC-based mRNA assays for early detection of HCC needs to be improved. Our streamlined HCC EV-based mRNA assay represents a promising noninvasive diagnostic solution for HCC early detection.

The streamlined HCC EV-based mRNA assay demonstrated high accuracy for differentiating HCC from noncancer (i.e., liver cirrhosis, chronic hepatitis, and healthy donors) and other cancer cohorts. While the at-risk cirrhotic population had very low EV Z scores across the board, three plasma samples (i.e., CLD12, CLD23, and CLD35) from chronic active HBV patients in the hepatitis cohort exhibited significantly higher signals in the heatmap, without any discernible HCC in imaging studies. Coincidently, two of these three patients (i.e., CLD12 and CLD23) had extremely high serum levels of HBV DNA of 3,752,532 and 20,900,652 IU mL$^{-1}$, respectively. In a longitudinal study of 3653 chronic hepatitis B patients with an elevated serum level of HBV DNA (>2000 IU mL$^{-1}$) at baseline, these patients were found to have an increased risk for subsequent development of HCC[59]. As such, continued surveillance of these three patients with high HCC EV Z scores may predict the development of HCC in the future.

We note that this study does have some limitations. The preclinical study was conducted using single cohorts for the HCC groups and at-risk liver cirrhotic group. The clinical reliability assessment was conducted in a small number of patients. In addition, the longitudinal follow-up was lacking. To further

progress toward practical HCC screening in at-risk populations, validation and testing cohorts as well as longitudinal follow-up will be required across all etiologies of HCC.

In conclusion, we have developed an HCC EV purification system (i.e., EV Click Chips) which allows for the digital scoring of HCC-specific mRNA transcripts. The resultant HCC EV Z Score was very specific, demonstrating accurate discrimination of HCC patients from human subjects without cancer and patients with other malignancies. Perhaps most importantly, our HCC EV-based mRNA assay displayed high sensitivity and superior performance in distinguishing early-stage HCC (BCLC Stage 0-A, or within Milan Criteria, or within UNOS DS Criteria) from at-risk liver cirrhotic patients, with the potential to allow for the detection of HCC in earlier stages when curative intent treatments are amenable. Our streamlined HCC EV-based mRNA assay holds great promise to significantly augment the ability of current HCC diagnostic modalities for early detection of HCC.

## Methods

**Fabrication of Tz-grafted SiNWS**. Our past experience in developing the NanoVilli EV Chip unveiled[24] that 10–15-μm long vertically aligned Si nanowires confer more effective surface area and sufficient mechanical robustness for EV capture. Hence, 10–15 μm Si nanowires (diameter = 100–200 nm) were introduced onto Tz-grafted SiNWS via a fabrication process combining photolithographic patterning and silver (Ag) nanoparticle-templated wet etching[60], offering ~30 times more surface area (in contrast to a flat substrate) for facilitating click chemistry-mediated EV capture. In accordance with the protocols published in our previous study[41], SiNWS were fabricated by combining the photolithographic patterning and Ag nanoparticle-templated wet etching[60]. In short, a p-type Si (100) wafer (Silicon Quest Int'l) was spin-coated with a thin film photoresist (AZ 5214, AZ Electronic Materials USA Corp.) using a resistivity of 10–20 Ω·cm. The Si wafer was then immersed into the etching solution containing HF (4.6 M, Sigma–Aldrich), $AgNO_3$ (0.2 M, Sigma–Aldrich) and deionized (DI) water after being exposed to ultraviolet light. Finally, the Ag nanoparticle-templates were removed by immersing the Si wafer into boiling aqua regia ($HCl/HNO_3$, 3:1 (v/v), Sigma–Aldrich) for 15 min. The SiNWS were then treated with acetone (≥99.5%, Sigma–Aldrich), followed by ethanol anhydrous (Sigma–Aldrich) wash. As shown in Supplementary Fig. 2, we introduced a disulfide linker to couple with the Tz motifs grafted on the chips by designing a three-step chemical modification procedure: (i) Silanization: The SiNWS were first immersed in a freshly prepared piranha solution ($H_2SO_4/H_2O_2$, 2:1 (v/v), Sigma–Aldrich) for 1 h, followed by rinsing with DI water and ethanol successively, three times. After drying under nitrogen flow, the resultant SiNWS were sealed in a vacuum desiccator for treatment with (3-mercaptopropyl) trimethoxysilane vapor (211.4 mg, 200 μL, Sigma–Aldrich) for 45 min to introduce thiol groups onto the SiNWS. (ii) Incorporation of disulfide bond: $OPSS-PEG-NH_2$ (0.30 mg, 3.8 mM, Nanocs Inc.) was incubated with freshly prepared HS-SiNWS in dimethyl sulfoxide (DMSO, 200 μL) solution for 2 h to introduce disulfide linkers with terminal amine groups. Then the amine-terminated SiNWS ($H_2N$-SiNWS) were rinsed with ethanol three times. (iii) To graft Tz motifs, the $H_2N$-SiNWS was incubated with Tz-sulfo-NHS ester (0.32 mg, 3.8 mM, Click Chemistry Tools Bioconjugate Technology Company) in PBS (200 μL, PH = 8.5) for 1 h. The resulting Tz-grafted SiNWS were rinsed with DI water three times. After drying under nitrogen flow, the Tz-grafted SiNWS were stored at −20 °C.

**Preparation of TCO-antibody conjugates**. Goat anti human EpCAM (R&D Systems, Inc., reconstitute at 0.2 mg/mL, dilute 400 to 2000 times in samples), goat anti human CD147 (R&D Systems, Inc., reconstitute at 0.5 mg/mL, dilute 1000–2000 times in samples), rabbit anti human ASGPR1 (LifeSpan BioSciences, Inc., 1 mg/ml, dilute 2000–10,000 times in samples), and sheep anti human GPC3 (R&D Systems, Inc., reconstitute at 0.2 mg/mL, dilute 200–2000 times in samples) were incubated with $TCO-PEG_4-NHS$ ester (0.5 mM, Click Chemistry Tools Bioconjugate Technology Company) in PBS using different mole ratios at room temperature for 30 min. The individual TCO-antibody conjugates were prepared freshly before their use.

**Cell line culture**. HepG2 and Hep3B cell lines were purchased from American Type Culture Collection and cultured in Eagle's Minimum Essential Medium with 10% fetal bovine serum (FBS), 1% GlutaMAX-I and 100 U $mL^{-1}$ penicillin-streptomycin (Thermo Fisher Scientific) in a humidified incubator with 5% $CO_2$. SNU387 cell line was purchased from American Type Culture Collection and cultured in RPMI-1640 Medium with 10% FBS, 1% GlutaMAX-I and 100 U $mL^{-1}$ penicillin-streptomycin in a humidified incubator with 5% $CO_2$.

**Artificial plasma sample preparation**. HepG2, Hep3B, SNU387 cells were cultured in 18 Nunc EasYDish dishes (145 cm², Thermo Fisher Scientific) for 72 h. Then the culture medium was switched to serum-free culture medium (Thermo Fisher Scientific) to starve the cells for 24–48 h. The serum-free culture medium incubated with cells was finally collected for EV isolation. After first centrifugation at 300 × g (4 °C) for 10 min to remove cells and cell debris, the supernatant was collected and transferred to new tubes and centrifuged at 2800 × g (4 °C) for 10 min to further eliminate the remaining cellular debris and large particles. The supernatant was carefully transferred to Ultra-Clear Tubes (38.5 mL, Beckman Coulter, Inc., USA), followed by ultracentrifugation using an Optima L-100 XP Ultracentrifuge (Beckman Coulter, Inc, USA) at 100,000 × g (4 °C) for 70 min. The EV pellets at the bottom of the tubes were carefully collected and resuspended in 200 μL fresh PBS. In control experiments, the EV pellets collected in 200 μL PBS were treated with RNase[61,62] at 37 °C for 30 min first, then PBS was added to wash and recollect the EV pellets by ultracentrifugation at 100,000 × g (4 °C) for 70 min. For the artificial plasma samples, each 10 μL aliquot of EV pellets was spiked into 90 μL healthy donors' plasma or cirrhotic patients' plasma.

**Characterization of HepG2 EVs**. For SEM characterization of HepG2 EVs, 10 μL pure HepG2 EVs in 100 μL PBS were run through the chips. The SiNWS were then cut to expose the cross sections of the silicon nanowire arrays. The severed SiNWS with captured HepG2 EVs were fixed in 4% PFA for 1 h, followed by sequential dehydration through 30, 50, 75, 85, 95, and 100% ethanol solutions for 10 min each. After overnight lyophilization, the samples were sputter-coated with gold at room temperature. The images were visualized and taken under a ZEISS Supra 40VP SEM at an accelerating voltage of 10 keV.

For TEM characterization of HepG2 EVs, 10 μL freshly harvested HepG2 EVs or purified HepG2 EVs were deposited on the 200-mesh formvar-carbon coated EM grids for 20 min, and then the grids were transferred (membrane side down) to a 100-μL drop of 4% PFA for 10 min. After water-drop washing three times, the grids were treated with 2% uranyl acetate for 5 min and excess fluid was blotted by filter paper. The grids air dried before TEM imaging by JEM1200-EX (JEOL USA Inc.) at 80 kV.

For the fluorescent labeling of captured EVs, 10 μL pure HepG2 EVs in 100 μL PBS were run through the chips. 1.2 μL PKH26 dye was added into 200 μL Diluent C and mixed continuously for 30 seconds by gentle pipetting. The severed SiNWS with captured HepG2 EVs were incubated with this PKH26 dye solution at room temperature for 10 min. The EV Click Chips after HCC EV capture and release were observed by fluorescence microscopy.

**EV click chips for HCC EV purification**. After chip assembly and leak testing according to our previously described protocols[24], the artificial plasma samples (100 μL) or clinical plasma samples (500 μL) incubated with TCO-antibodies were then injected into EV Click Chip microfluidic devices. For EV release, 100 μL DTT solution (50 mM) was injected into the EV Click Chips at 1.0 mL $h^{-1}$ and the released EVs were collected in 1.5 mL RNase-free Eppendorf tubes for subsequent RNA extraction.

**RNA extraction and RT-ddPCR**. The HCC EVs recovered from EV Click Chips were lysed by 700 μL QIAzol Lysis Reagent. RNA was extracted using a miRNeasy Micro Kit (Qiagen, USA) according to the manufacturer's instructions. Then the complementary DNA (cDNA) was synthesized using a Thermo Scientific Maxima H Minus Reverse Transcriptase Kit according to the manufacturer's instructions. For the optimization experiments, cDNA was tested for *SRY* transcripts and *C1orf101* transcripts using duplex ddPCR in one tube with two fluorescence filters (i.e., FAM and VIC). For the clinical samples, 10 μL of total cDNA (12 μL) was divided into five tubes to detect the 10 genes with two fluorescence filters in each tube. DdPCR experiments were performed on a QX200 system (Bio-Rad Laboratories, Inc.) according to the manufacturer's instructions. All primers and probes were purchased from Thermo Fisher Scientific and verified by cell lines and cell-line-derived EVs (See Supplementary Notes and Supplementary Fig. 8). Data were analyzed using the QuantaSoft™ software to quantify the corresponding copy numbers of gene transcripts detected in each assay.

**Enrollment of HCC patients and control cohorts**. All the participants in this study were enrolled between October 2016 and October 2019 at Ronald Reagan UCLA Medical Center and Cedars-Sinai Medical Center. All the participants are at least 18 years of age. Treatment-naïve HCC patients across all stages (n = 46) were enrolled in this study. HCC patients who had other malignant tumors or severe mental diseases were excluded. The control cohorts consisted of patients with liver cirrhosis (n = 26), chronic hepatitis B/C without liver cirrhosis (n = 25), other cancers with (n = 12) or without metastasis to liver (n = 26), and healthy donors (n = 23). A detailed description of each control cohort and clinical characteristics can be found in the Supplementary Information (Supplementary Tables 4–8). All patients and healthy donors provided written informed consent for this study according to the IRB protocol (IRB #14-000197) at UCLA and (IRB #00000066) at Cedars-Sinai Medical Center. None of the enrolled patients was a part of any clinical trial. Patient allocation to each of the cohorts was not random and was defined by their disease states.

**Clinical blood sample processing**. Peripheral venous blood samples were collected from fasting patients or healthy donors with written informed consent from each patient or healthy donor according to the institutional review board (IRB) protocols at UCLA and Cedars-Sinai Medical Center. Each 8.0 mL blood sample was collected in a BD Vacutainer glass tube (BD Medical, Fisher Cat. #02-684-26) with acid citrate dextrose. Samples were processed according to the manufacturer's protocol within 4 h of collection. The final plasma samples were collected for the HCC EV study after centrifugation at $10,000 \times g$ for 10 min. The plasma samples were aliquoted and stored in −80 °C refrigerators. Five hundred microliter plasma samples were then incubated with TCO-conjugated anti-EpCAM (250 ng), anti-ASGPR1 (125 ng) and anti-CD147 (125 ng) at room temperature for 30 min before being loaded into the EV Click Chips for the HCC EV purification. All plasma samples subjected to EV Click Chips and downstream RT-ddPCR assay underwent only one freeze-thaw cycle.

**Statistical analysis**. The EV recovery yields and purities are expressed as means ± SD. Significant differences between different groups were evaluated using one-way ANOVA. The 10-gene HCC EV Z Score, which represents the likelihood estimate of 10-gene activation, was computed from the RNA expression of the 10 genes using a weighted Z-score method[54] in R studio. After the mean centering of expression data across the samples, HCC EV Z Scores were computed by the error-weighted mean of the expression values of the 10 genes in a sample. We applied ROC curve to evaluate the diagnostic performance for each parameter using MedCalc software.

**Reporting summary**. Further information on research design is available in the Nature Research Reporting Summary linked to this article.

## Data availability

Raw data of Figs. 2c–f, h–j, 4b, 5 and Supplementary Figs. 5, 8d are provided as a Source Data file. All the other data supporting the findings of this study are available within the article and its supplementary information files and from the corresponding author upon reasonable request. Source data are provided with this paper.

## Code availability

The code for Z-score calculation used in this paper is provided in Supplementary Note 2.

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

## Acknowledgements

We acknowledge all the patients and healthy donors who participated in this study. This work was supported by National Institutes of Health grants (R21CA240887, R21CA216807, R21CA235340, R01CA218356, U01CA198900, R01CA253651, R01CA246304, and U01EB026421). N.S. gratefully acknowledges financial support from Collaborative Innovation Center of Suzhou Nano Science and Technology.

## Author contributions

Y.Z. and H.R.T. wrote the manuscript with input from all authors. Y.Z., H.R.T., V.G.A., and R.P. designed the research. N.S., S.J.C., and L.B. contributed cell culture and the cell-line-derived EV purification. N.S. and R.Y.Z. performed the optimization and RT-ddPCR experiments. N.S., P.J.C., P.C.T., Y.Y., J.P., N.N., S.H.H., S. Sadeghi, R.S.F., S. Saab, and R.W.B. contributed clinical samples collection. Y.T.L., J.J.W., and Y.Y. contributed clinical information collection. N.S., M.K., S.Y., D.M., D.E., J.W., and Y.Z. analyzed the data. N.S. and X.Z. contributed all figures in this study. R.K. conducted computational simulation. M.S., H.L., H.H.Y., A.P.H., J.D.Y., V.G.A., E.P., and R.P. revised the manuscript.

## Competing interests

The authors declare no competing interests.
