## [Peer Review File · Nature Communications]

Reviewers' comments:

Reviewer #1 (Remarks to the Author); expert on tumor extracellular vesicles:

Na Sun et al developed EV CLICK chips, a novel purification system for extracellular vesicles (EVs). The system makes use of click chemistry motifs on 1) packed silicon nanowired substrates and 2) HCC specific EV capture antibodies that irreversibly bind HCC specific EVs from blood plasma on the substrate. DTT treatment releases EVs from the substrate for downstream RT-ddPCR analysis for 10 HCC specific mRNA transcripts. The purification system developed looks promising but some clarification and additional experiments are required to demonstrate its full potential.

Major comments

The authors implement the quantification of SRY and C1 orf101 transcripts to assess the yield and the purity of the EV click chips. SRY transcripts are contributed by spiked male HepG2 EVs and C1 orf101 transcripts are contributed by EVs present in female donor plasma. Nevertheless it is not clear how well established and characterized the use of these transcripts for these purposes actually is. If available, this should be explained in the manuscript including references to the published work. If this transcript quantification has not been published before, some additional experimentation should be performed. The reason for this is that HepG2 cell derived EVs are obtained by ultracentrifugation at 100,000 g. Ultracentrifugation at 100,000 g does not only pellet EVs but also other extracellular particles containing RNA. What is the evidence that all SRY transcripts from the HepG2 spike are residing in EVs? And what is the evidence that all C1 orf101 transcripts are residing in EVs in the female donor plasma? In addition, what is the repeatability and/or reproducibility of this transcript quantification since this is rather crucial to define the recovery and purity?

The authors define purity as the specificity of the EV click chips for HCC EVs above non-HCC EVs. While in the EV research field, purity is often referred to as the specificity of the assay for EV compared to other extracellular particles such as lipoproteins or ribonucleoprotein complexes. It seems important to reconsider the definition of purity in this manuscript and to assess whether the EV click chip has issues with aspecific binding of those other extracellular particles besides non-HCC EVs.

Characterization of the different steps of the EV click chips is performed with HepG2 EVs spiked in PBS. But this is not representative of the EV click chips performed with blood plasma. It is crucial to repeat these characterization steps with HepG2 EVs spiked in blood plasma. This will reveal whether EV click chips are prone to aspecific binding of other extracellular particles. As a control in these experiments, the authors can spike blood plasma with HepG2 EVs that have been pre-treated with protease K (ie removal of surface proteins). The authors have also treated the HepG2 EVs with PKH26. It is not sure whether the fluorescence signal detected by confocal microscopy can be attributed to labeled EVs rather than aspecific binding of PKH26 micelles or other PKH26 labeled extracellular particles to the nanowires.

HCC specific EVs that are released from the EV click chips are further investigated by RT-ddPCR for 10 HCC specific mRNA transcripts. It seems that these mRNA transcripts have been selected based upon reference 33. However the authors from reference 33 studied these mRNAs in CTCs and not in EVs. Can the authors clarify what the evidence is that these 10 mRNAs are enriched in EVs? If this has not been published before, it seems crucial that HCC specific EVs that have been released by DTT from the EV click chips are pre-treated with protease K and RNase prior to perform ddPCR analysis to demonstrate that those mRNAs are enclosed in HCC-specific EVs.

Can the authors provide an explanation on the rather high HCC EV Z-scores in healthy donors in Figure 5a. From the supplementary table it seems that the average age of the healthy donor populations is much lower compared to the HCC population.

In line 275 the authors refer to a supplementary experiment to show the reproducibility of the ddPCR assay implemented in this study. Interesting would be, especially for a novel developed separation system for EVs, to demonstrate the reproducibility of the EV click chips.

Reviewer #2 (Remarks to the Author); expert on nanotechnology:

This is an interesting paper that uses a new technology with click chemistry for the selection of EVs for HCC management. The EV isolation technology is built from the work reported by this team for CTC isolation; SiNW are coupled to a PDMS chaotic mixer (this similarity does temper the level of novelty in this report). The clinical significance is high in that a liquid biopsy-based assay can be used for managing HCC and even to perform early detection. The authors were also able to perform mRNA expression profiling from EV-associated mRNA using ddPCR; the results in Figure 4 are very nice.

However, there are some issues that must be addressed in this manuscript before publication consideration can be given. For example, by using the EV number via NTA can this be used to distinguish disease states or must one use mRNA expression differences? This is important because significant amount of work and cost must be invested to perform RT-ddPCR.

Also, there is some comments made in the discussion section of the manuscript that are not really substantiated from data presented or referenced in the literature. For example, is the presented technology really the "best" for diagnosis of HCC, even early stage disease? There have been alternative EV isolation technologies that can be used for the early detection of different cancer-related diseases. In addition, there was no reference nor data that would indicate expression profiling from EVs is better than CTCs. Finally, the authors propose using click chemistry for catch and release; why not use the thermoresponsive materials that the authors have reported in previous papers for CTCs? This needs better justification.

Below are specific comments:

1. The click chemistry is interesting, but requires a reaction with plasma borne EVs (TCO) and another reaction with the surface of the SiNWs to position the Tz. What is the efficiency of these reactions and does unreacted Ab need to be removed prior to the assay or does the excess of TCO negate this need (see Figure 2b)?
2. It would be nice to understand the utility of the click approach used here versus the thermoresponsive coatings the author has used for CTC catch and release.
3. Like the idea of using the SiNWs because it will increase the available surface area of the device to increase its dynamic range, which is required for mRNA profiling. But, this architecture has been used in the past by the author for CTC isolation. This does temper the innovation somewhat.
4. The authors need to review or survey the numerous EV microchips that are already reported. Why does this SiNW chip with the click chemistry need to be used for this particular application?
5. For the data in figure 2g, the recovery seems to be flow rate independent from 0.2 to 1.0 mL/h. However, the recovery should be flow rate dependent because the EVs need to diffuse to the surface to interact with the Ab. Is the explanation due to the chaotic mixer? The authors need to discuss this. Bringing in results from the simulation (Figure S3) may help with this explanation.
6. What happened to the GBC-3 antigen? While it was listed in the text, there is no data to show why this was eliminated from the data in Figure 2f.
7. Authors state RNA degradation so high flow rates were used. However, the RNA is encapsulated within the membrane of the EV. The authors need to give a reference stating the fact that RNA degradation is seen in EVs.
8. The authors stated how the recovery was determined, but they also need to discuss how the

purity was determined as well.

9. How is the DTT removed prior to RT-ddPCR? Will not DTT deactivate the reverse transcriptase and polymerase? Does the SPE of the TRNA accomplish this?

10. How were the 10 genes selected? Did this come from Ref. 33?

11. Good idea to use properly designed primers due to the fact that most of the RNA is truncated with in the EVs.

12. For normalization of the ddPCR data in Figure 4, if normalize to the highest expression for each gene, was this done for each disease state or across all disease states for each gene? This needs to be clarified.

13. For the data in figure 4, it would be advisable to understand the amount of TRNA that was used in the RT-ddPCR results.

14. Comments for the discussion section of the manuscript.

a. There is probably no intact mRNA because most of the RNA is truncated before packaging into EVs. But, does the technology reported in here help to protect the mRNA? There were no experiments to demonstrate this.

b. There were no experiments to demonstrate the fact that the click chemistry was able to provide high recovery of even low expressing EVs in terms of the capture antigen (there was no data to show exactly how much antigen was expressed on the EVs).

c. There was specifically no experiments that showed irreversible immobilization resulting from using click chemistry.

d. The comment that the reported technology offers the most sensitive and specific capture of EVs is not founded - there is no review from the literature comparing other technologies.

e. For early stage HCC detection, agree that there may be a few CTCs, but the absolute particle number is not important, it is the amount of mRNA available for RT-ddPCR. For example, there are full length transcripts in CTCs and a higher amount of mRNA compared to what can be found in EVs, even if the absolute particle number is significantly less for the CTCs. Because the authors did not directly compare CTC vs. EV mRNA expression profiling for early stage HCC detection, this comment has not been validated.

Reviewer #3 (Remarks to the Author); expert on HCC early diagnosis:

The authors have described an elegant method for developing the EV chip capture development and early case-control validation study. They have found a high sensitivity and specificity in their validation of their technology in HCC patients vs controls. I have a few recommendations regarding this manuscript:

1. The author do not give an adequate sense of the reliability of the assay and reproducibility of the findings here. What was the sample run failure rate in the phase 2 validation? Was there reproducibility of the assays? Could the authors also give a sense of the ability to automate the steps outlines in sample preparation? Could this be feasibly rolled out in a clinical setting?

2. The authors list no limitations to this methodology in the discussion? Issues around reproducibility and ensuring that these findings are applicable across disease etiologies are important, especially since they are targeting a panel of tumor specific genes that may not be present in all etiologies of HCC. Future validation studies, including larger Phase 2 biomarker validation and Phase 3 biomarker validation should be discussed.

3. Please provide tumor size units for Supplemental Table 1. Also, were control samples with cirrhosis confirmed not to have HCC? Please provide details on how these patients were adjudicated as controls.

4. Please provide additional details for how the clinical samples were stored. Was there only one freeze-thaw cycle?

5. There is no racial information on the participants. Please provide this to get a sense in the heterogeneity of the samples.

Response to reviewer

Reviewer #1 (Remarks to the Author); expert on tumor extracellular vesicles:

Na Sun et al developed EV CLICK chips, a novel purification system for extracellular vesicles (EVs). The system makes use of click chemistry motifs on 1) packed silicon nanowired substrates and 2) HCC specific EV capture antibodies that irreversibly bind HCC specific EVs from blood plasma on the substrate. DTT treatment releases EVs from the substrate for downstream RT-ddPCR analysis for 10 HCC specific mRNA transcripts. The purification system developed looks promising but some clarification and additional experiments are required to demonstrate its full potential.

Major comments

Comment 1: The authors implement the quantification of SRY and C1orf101 transcripts to assess the yield and the purity of the EV click chips. SRY transcripts are contributed by spiked male HepG2 EVs and C1orf101 transcripts are contributed by EVs present in female donor plasma. Nevertheless it is not clear how well established and characterized the use of these transcripts for these purposes actually is. If available, this should be explained in the manuscript including references to the published work. If this transcript quantification has not been published before, some additional experimentation should be performed. The reason for this is that HepG2 cell derived EVs are obtained by ultracentrifugation at 100,000 g. Ultracentrifugation at 100,000 g does not only pellet EVs but also other extracellular particles containing RNA. What is the evidence that all SRY transcripts from the HepG2 spike are residing in EVs? And what is the evidence that all C1orf101 transcripts are residing in EVs in the female donor plasma? In addition, what is the repeatability and/or reproducibility of this transcript quantification since this is rather crucial to define the recovery and purity?

Response 1: We thank the reviewer for his/her comments. The quantification of SRY and C1orf101 transcripts for assessing the yield and purity of EV click chips hasn't been published before. As we showed in **Figure 2a**, this quantitative method can be used when the HepG2 cell-derived EVs were spiked into an opposite-sex healthy donor plasma used as the artificial plasma samples. we have explained this method in detail in **Pages 8-9, Lines 150-165**, and in **Supplementary Information 4 and Supplementary Figure S4b-c**.

The reviewer expressed concern that the SRY mRNA transcripts in our artificial plasma sample could come from other extracellular particles containing RNA (e.g., circulating ribonucleoprotein). We feel our approach mitigates this problem for several reasons: 1) EV Click Chips adopt the covalent chemistry-mediated EV purification approach. We are confident that the combined use of click chemistry-mediated EV "capture" and disulfide cleavage-driven EV "release" confers dramatically improved sensitivity and specificity to achieve targeted enrichment of HCC cell line-derived EVs with minimal level of nonspecific trapping of other extracellular particles. 2) To further address the reviewer's concern, we treated HCC EVs (collected from three HCC cell lines, i.e., HepG2, SNU387, and Hep3B) with ribonucleases (RNase) to remove circulating ribonucleoprotein and non-protected RNA. The RNase-treated HCC EVs were spiked into two types of plasma samples (collected from either healthy donors or liver cirrhotic patients) to give six different artificial plasma samples. The original HCC EV purification studied summarized in Figure 2j were repeated using these six artificial plasma samples. As shown in **Supplementary Table S1 (Lines 280-282, Page S18)** and **Supplementary Table S3 (Lines 292-297, Page S19)**, overall, EV Click Chips achieved a

similar degree of recovery yields ranging from 74.4% to 86.7% and recover purities ranging from 82.4% to 99.1%. These results further support that EV Click Chips is capable of specifically enrich HCC EVs without nonspecific trapping of other extracellular particles containing RNA. Together with the original Figure 2j, transcript quantification presented in this manuscript were all repeated 3 times using 3 cell lines w/wo RNase treatment. The reproducibility of this transcript quantification is excellent, and the results were also shown in **Supplementary Table S3**.

Supplementary Table S1. The results of reproducibility study of C1orf101/SRY transcript quantification.

samples	Ratio of C1orf101 transcripts to SRY transcripts in original cell line EVs (C1orf101/SRY)					Intra CV (%)
	Test 1	Test 2	Test 3	Test 4	Mean	
HepG2	2.04 (253/124)	1.97 (1770/894)	1.97 (10020/5080)	1.83 (16680/9120)	1.95	4.51
Male healthy donor 01	1.0 (2.6/2.6)	1.18 (10.6/9)	1.11 (18/16.2)	N/A	1.10	8.27
Male liver cirrhotic patient01	1.0 (2.8/2.8)	1.0 (1.4/1.4)	N/A	N/A	1.0	0

Supplementary Table S3. The results of reproducibility study of the quantitative method for assessing the performance of EV Click Chips using artificial samples of different cell line-derived EVs spiked into different background plasmas.

without RNase	Recovery Yield (%)						Recovery Purity (%)					
	Test1	Test2	Test3	Mean	STDEV	%CV	Test1	Test2	Test3	Mean	STDEV	%CV
HepG2/HD	82.4	82.4	84.3	83.0	1.1	1.29	91.3	83.7	89.6	88.2	3.3	2.66
SNU387/HD	88.9	88.7	90.8	89.5	1.1	1.33	99.1	98.8	99.1	99.0	0.1	0.14
Hep3B/HD	94.6	90.6	92.3	92.5	2.0	2.17	97.6	96.5	95.2	96.4	1.0	1.02
HepG2/CLD	87.1	96.5	91.8	91.8	4.7	0.78	96.5	89.8	92.9	93.1	2.8	3.99
SNU387/CLD	82.3	81.0	83.4	82.2	1.2	1.46	99.9	98.8	99.9	99.5	0.5	0.47
Hep3B/CLD	81.2	84.2	81.5	82.3	1.7	2.01	92.4	97.5	92.8	94.2	2.3	2.46
with RNase	Recovery Yield (%)						Recovery Purity (%)					
	Test1	Test2	Test3	Mean	STDEV	%CV	Test1	Test2	Test3	Mean	STDEV	%CV
HepG2/HD	81.3	85.8	82.3	83.1	2.4	0.78	83.7	86.3	87.9	86.0	1.7	0.64
SNU387/HD	78.7	83.5	83.1	81.8	2.7	3.18	99.7	98.3	97.7	98.6	0.9	0.85
Hep3B/HD	90.6	89.5	83.1	87.7	4.1	6.65	97.9	96.7	97.7	97.5	0.5	0.54
HepG2/CLD	92.9	97.5	90.2	93.5	3.7	1.49	82.8	83.7	81.2	82.6	1.0	1.23
SNU387/CLD	69.5	77.4	76.3	74.4	4.2	5.75	99.5	99.7	98.2	99.1	0.7	0.67
Hep3B/CLD	75.4	87.3	74.0	78.9	7.3	9.26	93.0	94.3	87.1	91.5	3.1	3.43

Source data are provided as a Source Data file.

Comment 2: The authors define purity as the specificity of the EV click chips for HCC EVs above non-HCC EVs. While in the EV research field, purity is often referred to as the specificity of the assay for EV compared to other extracellular particles such as lipoproteins or ribonucleoprotein complexes. It seems important to reconsider the definition of purity in this manuscript and to assess whether the EV click chip has issues with aspecific binding of those other extracellular particles besides non-HCC EVs.

Response 2: We thank the reviewer for his/her comments. We agree that in the EV research field EV purity is often referred to as the specificity of the assay for EV compared to other extracellular particles. To avoid confusing, we have redefined “recovery purity” as the specificity of HCC EVs compared to non-HCC EVs, and revised the definition throughout the manuscript. Regarding to the issue of non-specific trapping of those other extracellular particles besides non-HCC EVs in EV Click Chip, please refer to **Response 1**.

Comment 3: Characterization of the different steps of the EV click chips is performed with HepG2 EVs spiked in PBS. But this is not representative of the EV click chips performed with blood plasma. It is crucial to repeat these characterization steps with HepG2 EVs spiked in blood plasma. This will reveal whether EV click chips are prone to aspecific binding of other extracellular particles. As a control in these experiments, the authors can spike blood plasma with HepG2 EVs that have been pre-treated with protease K (ie removal of surface proteins). The authors have also treated the HepG2 EVs with PKH26. It is not sure whether the fluorescence signal detected by confocal microscopy can be attributed to labeled EVs rather than aspecific binding of PKH26 micelles or other PKH26 labeled extracellular particles to the nanowires.

Response 3: We thank the reviewer for his/her comments. To address the reviewer’s questions, we have conducted two parallel characterization studies. In the first study, RNase-treated HepG2 EVs were first labeled with PKH26 dye and then spiked into healthy donors’ plasma samples. The resulting artificial samples were subjected to the HCC EV capture/release workflow shown in **Supplementary Figure S7a**. Again, fluorescent microscopy was employed for tracking the purification (capture/release) process of RNase-treated HepG2 EVs in EV Click Chips. The result is very similar to that observed for HepG2 EVs in PBS (Figure3). In the second study, RNase-treated HepG2 EVs were first exposed to protease K, followed by PKH26 dye labeling. The resulting HepG2 EVs were then spiked into a healthy donors’ plasma samples, and these artificial samples were subjected to the HCC EV capture/release workflow shown in **Supplementary Figure S7b**. Fluorescent microscopy imaging revealed negligible fluorescent signals, suggesting that the click chemistry-mediated capture was not able to immobilize PKH26-labeled HepG2 EVs (with removal of surface proteins) on SiNWS. These results indicated that the enzymatic removal of HCC-associated surface markers on HCC EVs led to the failure of EV capture in EV Click Chips.

For the concerns on EV labeling by PKH26 dye (a widely-used EV labeling dye), we washed the PKH26-labeled EVs immobilized on the chips with PBS prior to the microscopy imaging to avoid non-specific trapping of free PKH26 dye. Moreover, we conducted a control experiment, where PKH26 Dye in healthy donors’ plasma (without HCC EVs) was run through the chips. After washing with PBS, no fluorescent signals were observed on the EV Click Chips. Since PKH26 dye can only stain lipid bilayer membrane structured particles, other particles without membranes cannot be stained in this case. These results demonstrated that the fluorescent signals detected under microscopy in our studies can be attributed to labeled EVs rather than non-specific binding of PKH26 micelles or other PKH26 labeled extracellular particles to the nanowires. We have

summarized these results in **Supplementary Figure S7c**.

Supplementary Figure S7. Tracking the purification (capture/release) process of (a) HepG2 EVs treated with RNase and (b) HepG2 EVs treated with RNase & protease K in EV Click Chips using fluorescent microscopy. (c) Tracking the capture process of EVs in healthy donor plasma. Size distribution of HepG2 EVs in solution measured by DLS (d) before and (e) after purification by EV Click Chips. (f) Immunogold labeling with anti-CD63 to verify the purified HepG2 EVs from EV Click Chips (10 nm gold particles)

Comment 4: HCC specific EVs that are released from the EV click chips are further investigated by RT-ddPCR for 10 HCC specific mRNA transcripts. It seems that these mRNA transcripts have been selected based upon reference 33. However, the authors from reference 33 studied these mRNAs in CTCs and not in EVs. Can the authors clarify what the evidence is that these 10 mRNAs are enriched in EVs? If this has not been published before, it seems crucial that HCC specific EVs that have been released by DTT from the EV click chips are pre-treated with protease K and RNase prior to perform ddPCR analysis to demonstrate that those mRNAs are enclosed in HCC-specific EVs.

Response 4: As we indicated in lines 271 and 289, Page 15, these mRNA transcripts were selected

based upon reference 39. First, we confirmed that these 10 mRNA markers are detectable in EVs by checking the publicly available EV databases, such as ExoCarta, Vesiclepedia, and exoRBase. Then, we validated mRNA expression using HepG2 EVs, with the results in **Supplementary Figure S8** demonstrating that these mRNAs are enriched in EVs. Moreover, the 10-gene signatures in recovered HCC EVs obtained from 158 clinical samples also confirmed that these 10 transcripts are enriched in EVs. Additionally, following the reviewer #1's suggestion, we have conducted experiments using EVs pre-treated with RNase prior to EV capture (before ddPCR analysis). More details can be found in our response to Comment 1 from Reviewer 1. Taken together, the 10 mRNA markers detected by ddPCR are from EVs enriched by EV click chips.

Comment 5: Can the authors provide an explanation on the rather high HCC EV Z-scores in healthy donors in Figure 5a. From the supplementary table it seems that the average age of the healthy donor populations is much lower compared to the HCC population.

Response 5: The reviewer raised a great question. We would like to explain why higher HCC EV Z-scores were obtained in healthy donors (**Figure 5a**). First, expression levels and variance of ALB gene are higher in healthy donors compared to cirrhotic group. However, rest of the genes were almost not expressed or very low in both healthy donors and cirrhotic groups as you can see in **Figure 4b**. Due to this, higher HCC EV Z-scores in healthy donors were observed than ones in the liver cirrhotic group. This lower expression of ABL gene in cirrhotic group can be explained by the fact that patients with cirrhosis have impaired hepatocellular function and reduced albumin synthesis, which can reach a 60-80% reduction in advanced cirrhosis (reference: Current indications for the use of albumin in the treatment of cirrhosis. *Ann Hepatol*, 10 (2011), pp. 15-20). Second, higher HCC EV Z-scores were observed in healthy donors than the other cancer group. This is because the HCC-specific 10 genes will not read out the signals from other cancer-derived EVs, which dominate the EVs purified from the EV Click Chip. Therefore, the ALB signals from the liver-derived EVs will be diluted in these other cancer samples.

With regards to the age of the healthy donor populations, 5 healthy donor samples at the age of 20s-30s (HD04, HD17, HD18, HD19, HD20) were replaced by 5 more samples of healthy donors (HD24, HD25, HD26, HD27, HD28) with ages matched to HCC populations. All data in **Figure 4**, **Figure 5** and **Supplementary Table S7** have been updated accordingly.

Comment 6: In line 275 the authors refer to a supplementary experiment to show the reproducibility of the ddPCR assay implemented in this study. Interesting would be, especially for a novel developed separation system for EVs, to demonstrate the reproducibility of the EV click chips.

Response 6: We thank the reviewer for his/her comments. We have performed additional reproducibility study using artificial samples as well as 5 clinical samples. The results are summarized in the **Supplementary Table S2** and **Supplementary Figure S8d**.

For the artificial samples, the reproducibility was measured by calculating the percent coefficient of variation (%CV) for recovery yields. Intra-assay variability was measured for one operator who performed three tests on one day, whereas inter-assay variability was measured across three operators who performed five assay runs total (one run per day), with each run consisting of three tests (15 chips total). We have summarized the updated data in revised **Figure 2h**, and the percent coefficient of variation (%CV) was listed in **Supplementary Table S2**.

Supplementary Table S2. The results of reproducibility study of SRY transcript quantification for assessing the recovery yield of EV Click Chips using artificial samples spiked with different concentrations of EVs.

Run Number	Recovery Yield (%)				Intra CV (%)
	Test 1	Test 2	Test 3	Mean	
1	73.3	90	93.7	85.7	12.65
2	95.4	81.8	93.8	90.3	8.22
3	87.5	93.8	80.2	87.2	7.83
4	88.5	93.9	80.1	87.5	7.91
5	81.6	81.8	80.1	81.2	1.12
Inter CV (%)				3.88	

Source data are provided as a Source Data file.

For clinical samples, we tested 5 HCC patient plasma samples. Each HCC patient plasma was split into 3 samples for independent analysis by HCC EV purification and HCC EV-specific gene profiling. The intra-class correlation coefficient (ICC) of 10 HCC EV-mRNA markers for each patient was calculated based upon three repeated measurements (ICC= 0.93; 95% CI, 0.89-0.96). The results were summarized in **Supplementary Figure S8d**.

Supplementary Figure S8. Validation of primers and probes for the 10-gene panel using multiplex-droplet digital PCR (ddPCR). (a) Signals of the 10 genes in HepG2 cells (positive control) and healthy donor (HD) white blood cells (WBCs) (negative control). (b) Signals of the 10 genes in the original HepG2-derived EVs, EVs recovered by Click Chips using artificial plasma sample (EV-sample 1, HepG2-derived EVs were spiked into a healthy donor's plasma), and artificial at-risk plasma sample (EV-sample 2, HepG2-derived EVs were spiked into a liver cirrhotic patient's plasma). (c) Schematic illustrating the design and gene assignments in multiplex-ddPCR for the 10-gene panel. (d) The reproducibility study of HCC EV-based mRNA assay using 5 HCC patients' samples. Source data are provided as a Source Data file. (e) Heatmap of 15 samples from 5 HCC patients' samples for reproducibility study.

Reviewer #2 (Remarks to the Author); expert on nanotechnology:

This is an interesting paper that uses a new technology with click chemistry for the selection of EVs for HCC management. The EV isolation technology is built from the work reported by this team for CTC isolation; SiNW are coupled to a PDMS chaotic mixer (this similarity does temper the level of novelty in this report). The clinical significance is high in that a liquid biopsy-based assay can be used for managing HCC and even to perform early detection. The authors were also able to perform mRNA expression profiling from EV-associated mRNA using ddPCR; the results in Figure 4 are very nice.

However, there are some issues that must be addressed in this manuscript before publication consideration can be given. For example, by using the EV number via NTA can this be used to distinguish disease states or must one use mRNA expression differences? This is important because significant amount of work and cost must be invested to perform RT-ddPCR.

Response A: We thank the reviewer for his/her comments. We appreciate the fact that NTA (Nanoparticle Tracking Analysis) technique is used widely in the EV field. NTA relies on light scattering to measure EV size and number. However, some concerns have been raised regarding variability and reproducibility of the NTA measurements. As an alternative solution, we characterized EV size by SEM and TEM, and quantified purified EVs by measuring the mRNA markers in the EVs (using RT-ddPCR) before and after purification by EV Click Chips. In addition, NTA cannot distinguish whether the purified EVs are tumor-derived or not. As we discussed in the manuscript, “The combined use of a multimarker antibody cocktail and EV Click Chips could possibly lead to recovering EVs which are not of HCC origin. For example, anti-EpCAM could capture EVs from other epithelial tissues. To address this concern, we adopted the RT-ddPCR assay capable of quantifying 10 HCC-specific genes as a downstream readout for the purified HCC EVs.” Moreover, EpCAM is known to be highly expressed in most epithelial carcinomas. In this case, HCC can’t be distinguished from other cancers when using only EV number. More specific downstream characterization solution, a 10 HCC-specific gene panel used in this manuscript, helps to solve this problem, as shown in **Figure 4** and **5**.

Also, there is some comments made in the discussion section of the manuscript that are not really substantiated from data presented or referenced in the literature. For example, is the presented technology really the “best” for diagnosis of HCC, even early stage disease? There have been alternative EV isolation technologies that can be used for the early detection of different cancer-related diseases. In addition, there was no reference nor data that would indicate expression profiling from EVs is better than CTCs. Finally, the authors propose using click chemistry for catch and release; why not use the thermoresponsive materials that the authors have reported in previous papers for CTCs? This needs better justification.

Response B: To avoid potential confusion in the discussion section, we have revised the language by deleting “the most”. We also revised our discussion on CTC’s diagnostic potential to no longer compare CTCs and EVs. The aim of the present study is to explore a novel HCC EV purification system and its downstream mRNA profiling for HCC early diagnosis.

We appreciate the reviewer’s question regarding the use of Thermoresponsive NanoVelcro Chip (ACS Nano 2015) for purification of HCC EVs. We actually examined the feasibility of capturing HCC EVs using the Thermoresponsive NanoVelcro Chips and discovered two challenges. First, Thermoresponsive NanoVelcro Chip is based on antibody-mediated EV capture which exhibited

limited capture performance. This drove us to develop Click Chemistry-mediated EV capture to improve the performance. Second, temperature-drive release (at 4°C) of EVs in the Thermoresponsive NanoVelcro Chips resulted in EV rupture, leading to poor EV recovery performance. Overall, Thermoresponsive NanoVelcro Chips are more suitable for the purpose of CTC purification. The EV Click Chips reported in this manuscript serve better for the purpose of EV purification.

Below are specific comments:

Comment 1: The click chemistry is interesting, but requires a reaction with plasma borne EVs (TCO) and another reaction with the surface of the SiNWs to position the Tz. What is the efficiency of these reactions and does unreacted Ab need to be removed prior to the assay or does the excess of TCO negate this need (see Figure 2b)?

Response 1: We did NOT remove unbound TCO-grafted antibodies since there were more than 10 times as many Tz-motifs on SiNWs than TCO motifs on both TCO-grafted HCC EVs and free TCO-grafted antibodies. Therefore, all TCO-grafted HCC EVs and free TCO-grafted antibodies could be immobilized onto the SiNWs.

Comment 2: It would be nice to understand the utility of the click approach used here versus the thermoresponsive coatings the author has used for CTC catch and release.

Response 2: Please refer to **response B** above.

Comment 3: Like the idea of using the SiNWs because it will increase the available surface area of the device to increase its dynamic range, which is required for mRNA profiling. But, this architecture has been used in the past by the author for CTC isolation. This does temper the innovation somewhat.

Response 3: We thank the reviewer for these positive comments. We agree with the reviewer's perspective on the novelty of the device architecture. We would like to note that originality of this manuscript stems from the following two aspects: 1) synergistic integration of four powerful approaches (including covalent chemistry-mediated EV capture/release, multimarker antibody cocktails, nanostructured substrates, and a microfluidic chaotic mixer) to achieve purification of HCC EVs, and 2) combining EV Click Chip with RT-ddPCR for early detection of HCC from at-risk cirrhotic patients.

Comment 4: The authors need to review or survey the numerous EV microchips that are already reported. Why does this SiNW chip with the click chemistry need to be used for this particular application?

Response 4: We appreciate this helpful suggestion. In response, we have identified and cited more work with EV microchips in **Line 68, Page 4**. Immunoaffinity-based EV capture approaches, which are driven by the dynamic binding between a pair of antigens (on EVs) and antibodies (on the substrates), often suffer from poor EV capture performance and high background. We attempted to address these issues by introducing click-chemistry-mediated EV capture. Among different categories of click chemistry reactions, we selected the inverse-electron-demand Diels-Alder cycloaddition between Tz and TCO motifs (a rate constant of $10^4 \text{ M}^{-1}\cdot\text{s}^{-1}$), considering their balanced chemical properties between reactivity and stability. The ligation between Tz-grafted SiNWs and TCO-grafted EVs is rapid, specific, irreversible, and insensitive to biomolecules, water, and oxygen, leading to immobilization of the EVs with improved capture efficiency and reduced

nonspecific trapping of particles in the background.

Comment 5: For the data in figure 2g, the recovery seems to be flow rate independent from 0.2 to 1.0 mL/h. However, the recovery should be flow rate dependent because the EVs need to diffuse to the surface to interact with the Ab. Is the explanation due to the chaotic mixer? The authors need to discuss this. Bringing in results from the simulation (Figure S3) may help with this explanation.

Response 5: We thank the reviewer for pointing out this matter. In this manuscript, the TCO-conjugated antibody agents were incubated with EVs in the artificial plasma samples before being subjected to the EV click chips. The EV recovery yield is flow-rate dependent as shown in **Figure 2g**. To elucidate how the cooperation of the chaotic mixer and SiNWS facilitates the Click Chemistry-mediated EV capture, we referred to the computational simulation results (**Supplementary Figure S3**) based on computational fluid dynamics (CFD) and dissipative particle dynamics (DPD) models. First, the CFD model simulated the trajectories (**Supplementary Figure S3c/d**) of the flow-through EVs introduced by the embedded herringbone patterns in the chaotic mixer, suggesting that EVs can be effectively introduced into $<10\ \mu\text{m}$ -thick boundary layers on SiNWS, where the EVs can then diffuse into SiNWS via Brownian motion for Click Chemistry-mediated EV capture. The DPD model was then employed to depict Brownian motion of EVs into the Si nanowires on SiNWS to simulate their vertical distribution (**Supplementary Figure S3f-right and h**) of EVs along the Si nanowires. The simulated results show that the DPD simulated ($n=48$) EV distribution is consistent with our optimized experimental data ($n=108$) as shown in **Supplementary Figure S3e, f-left, and g**. According to our simulation model, when flow rate is high ($> 2\ \text{mL/h}$), EVs exhibit limited time in the boundary layers on SiNWS. As a result, there is insufficient time for EVs to diffuse into SiNWS (via Brownian motion) to achieve the desired performance via Click Chemistry-mediated EV capture.

Comment 6: What happened to the GPC-3 antigen? While it was listed in the text, there is no data to show why this was eliminated from the data in Figure 2f.

Response 6: The experimental data for anti-GPC3 as an additional EV capture agent was shown in **Supplementary Figure S5**. The EV recovery yield is not satisfied in the presence of anti-GPC3.

In addition, The HepG2 EV recovery yields were compared among different groups of single antibodies and antibody cocktails, and anti-EpCAM & anti-ASGPR & anti-CD147 was selected as the optimal multi-marker cocktail for capturing HCC EVs.

Comment 7: Authors state RNA degradation so high flow rates were used. However, the RNA is encapsulated within the membrane of the EV. The authors need to give a reference stating the fact that RNA degradation is seen in EVs.

Response 7: We have revised the description in **Page 12, Lines 212-213**. “To allow for a faster turnaround time for clinical samples, the flow rate of $1.0\ \text{mL h}^{-1}$ was selected”.

Comment 8: The authors stated how the recovery was determined, but they also need to discuss how the purity was determined as well.

Response 8: In this manuscript, the “recovery purity” was defined as the specificity of HCC EVs compared to non-HCC EVs. We discussed how the purity was determined in **Page 8-9, Line 154-162**. “In order to obtain the purity of the EVs recovered by EV Click Chips, we first measured the intrinsic ratios between C1orf101 and SRY transcripts in aliquoted HepG2 EVs across a wide range of concentrations. As shown in **Supplementary Figure S4a**, the ratios between C1orf101 and SRY transcripts in HepG2 EVs exhibited a consistent linear correlation ($y = 1.95 x$, $R^2 = 0.999$). With the

C1orf101-to-SRY ratio determined as 1.95, we then calculate the purity of the HepG2 EVs harvested from EV Click Chips as the ratio of the recovered SRY transcripts (contributed by recovered HepG2 EVs only) to the C1orf101 transcripts (contributed by both recovered HepG2 EVs and the non-specifically captured background plasma-derived EVs, denoted as C1orf101 gene_{rec-EV}) using the following equation:”

$$\text{HCC EV recover purity} = \frac{\text{SRY transcripts}_{\text{rec-EV}}}{\text{C1orf101 transcripts}_{\text{rec-EV}}} \times 1.95^* \quad (2)$$

*1.95 is specific to HepG2 EVs.

Comment 9: How is the DTT removed prior to RT-ddPCR? Will not DTT deactivate the reverse transcriptase and polymerase? Does the SPE of the TRNA accomplish this?

Response 9: The DTT was removed during the RNA extraction process. We use Qiagen miRNeasy Micro Kit for RNA extraction. The sample goes through lysis buffer, chloroform and ethanol. Since DTT is highly soluble in ethanol and chloroform, it will be dissolved and removed before RT and PCR.

Comment 10: How were the 10 genes selected? Did this come from Ref. 33?

Response 10: The selection of 10 HCC-specific genes (AFP, GPC3, AHSB, ALB, APOH, FABP1, FGB, FGG, RBP4, and TF) was based on the panel for circulating tumor cell (CTC) detection published by Kalinich *et al.* at the Massachusetts General Hospital for HCC early detection. The expression of these genes is well-validated and well-suited for our HCC-specific extracellular vesicle (EV) digital scoring assay. Additional information relevant to this issue is available in our reply (**Response 4**) to **Reviewer 1**.

Comment 11: Good idea to use properly designed primers due to the fact that most of the RNA is truncated with in the EVs.

Response 11: We appreciate the positive comments on the design for HCC EV-specific RT-ddPCR protocols.

Comment 12: For normalization of the ddPCR data in Figure 4, if normalize to the highest expression for each gene, was this done for each disease state or across all disease states for each gene? This needs to be clarified.

Response 12: The normalization was done for all disease states for each gene. We have clarified the normalization method in **Lines 296-297, Page 16**.

Comment 13: For the data in figure 4, it would be advisable to understand the amount of TRNA that was used in the RT-ddPCR results.

Response 13: The amount of RNA recovered from each chip that was used in the following RT-ddPCR analysis ranging from 0.04 ng/μL to 12.0 ng/μL (9 μL RNA solution could be obtained after RNA extraction). We have added RNA amount for 16 clinical samples in the manuscript **Lines 283-284, Page 15** and summarize the RNA concentrations in a **Supplementary Table S9**.

Supplementary Table S9. RNA concentrations of patients' samples used in the following RT-ddPCR analysis.

Sample ID	RNA concentration (ng/ul)
HCC01	2.63
HCC03	0.53
HCC05	12.00
HCC08	5.37
HCC14	1.77
HCC19	1.08
HCC21	0.10
HCC27	1.96
HCC28	0.42
HCC31	0.04
HCC32	0.19
HCC33	2.25
CLD05	0.82
CLD06	3.00
HD02	0.79

Comment 14: Comments for the discussion section of the manuscript.

a. There is probably no intact mRNA because most of the RNA is truncated before packaging into EVs. But, does the technology reported in here help to protect the mRNA? There were no experiments to demonstrate this.

Response a: We have revised our discussion on this point in **Lines 288-291, Page 15**. It was reported that EVs contain full-length but sometimes fragmented RNA (*Reference 50*). Our purification method is mild, therefore, the membrane of EVs remains intact, thus the mRNA cargo was protected by the membrane.

b. There were no experiments to demonstrate the fact that the click chemistry was able to provide high recovery of even low expressing EVs in terms of the capture antigen (there was no data to show exactly how much antigen was expressed on the EVs).

Response b: We have revised our discussion on this point in **Lines 355-358, Page 20**. We have demonstrated that the performance of click-chemistry-mediated EV capture is superior to the immunoaffinity-based EV capture approaches on the same Nanostructured device (Figure 2i), which are driven by the dynamic binding between a pair of antigens (on EVs) and antibodies (on the substrates).

c. There was specifically no experiments that showed irreversible immobilization resulting from using click chemistry.

Response c: We have revised our discussion on this point in **Line 363-364, Page 20**. In fact, it's the click chemistry-mediated reactions that are irreversible (*reference 37*).

d. The comment that the reported technology offers the most sensitive and specific capture of EVs is not founded - there is no review from the literature comparing other technologies.

Response d: We have revised our description on this point in **Lines 377-378, Page 21**. Our technology offers a sensitive and specific technology for capturing HCC EVs.

e. For early stage HCC detection, agree that there may be a few CTCs, but the absolute particle number is not important, it is the amount of mRNA available for RT-ddPCR. For example, there are full length transcripts in CTCs and a higher amount of mRNA compared to what can be found in EVs, even if the absolute particle number is significantly less for the CTCs. Because the authors did not directly compare CTC vs. EV mRNA expression profiling for early stage HCC detection, this comment has not been validated.

Response e: We appreciate this helpful suggestion and have revised the discussion accordingly (**Lines 409-410, Page 22**).

Reviewer #3 (Remarks to the Author); expert on HCC early diagnosis:

The authors have described an elegant method for developing the EV chip capture development and early case-control validation study. They have found a high sensitivity and specificity in their validation of their technology in HCC patients vs controls. I have a few recommendations regarding this manuscript:

Comment 1a: The author do not give an adequate sense of the reliability of the assay and reproducibility of the findings here.

Response 1a:

We thank the reviewer for his/her comments. We have performed the reproducibility studies using artificial samples as well as 5 clinical samples. The results are summarized in **Supplementary Table S2** and **Supplementary Figure S8d**.

For the artificial samples, the reproducibility was measured by calculating the percent coefficient of variation (%CV) for Recovery Yields. Intra-assay variability was measured for one operator who performed three tests on one day, whereas inter-assay variability was measured across three operators who performed five assay runs total (one run per day), with each run consisting of three tests (15 chips total). We have summarized the updated data in revised **Figure 2h**, and the percent coefficient of variation (%CV) was listed in **Supplementary Table S2**.

Supplementary Table S2. The results of reproducibility study of SRY transcript quantification for assessing the recovery yield of EV Click Chips using artificial samples spiked with different concentrations of EVs.

Run Number	Recovery Yield (%)				Intra CV (%)
	Test 1	Test 2	Test 3	Mean	
1	73.3	90	93.7	85.7	12.65
2	95.4	81.8	93.8	90.3	8.22
3	87.5	93.8	80.2	87.2	7.83
4	88.5	93.9	80.1	87.5	7.91
5	81.6	81.8	80.1	81.2	1.12
Inter CV (%)				3.88	

Source data are provided as a Source Data file.

For clinical samples, we tested 5 HCC patient plasma samples, each HCC patient plasma was split into 3 samples for independent analysis by HCC EV purification and HCC EV-specific gene profiling. The intra-class correlation coefficient (ICC) of 10 HCC EV-mRNA markers for each patient was calculated based upon three repeated measurements (ICC= 0.93; 95% CI, 0.89-0.96). The results were summarized in **Supplementary Figure S8d**.

Supplementary Figure S8. Validation of primers and probes for the 10-gene panel using multiplex-droplet digital PCR (ddPCR). (a) Signals of the 10 genes in HepG2 cells (positive control) and healthy donor (HD) white blood cells (WBCs) (negative control). (b) Signals of the 10 genes in the original HepG2-derived EVs, EVs recovered by Click Chips using artificial plasma sample (EV-sample 1, HepG2-derived EVs were spiked into a healthy donor’s plasma), and artificial at-risk plasma sample (EV-sample 2, HepG2-derived EVs were spiked into a liver cirrhotic patient’s plasma). (c) Schematic illustrating the design and gene assignments in multiplex-ddPCR for the 10-gene panel. (d) The reproducibility study of HCC EV-based mRNA assay using 5 HCC patients’ samples. Source data are provided as a Source Data file. (e) Heatmap of 15 samples from 5 HCC patients for reproducibility study.

Comment 1b. What was the sample run failure rate in the phase 2 validation?

Response 1b: In the Phase 2 “Clinical Assay and Validation (Clinical assay detects established disease)” of this study, of the 158 samples in preliminary study, 1 sample was failed during the analysis, thus the sample run failure rate is: $1/158=0.63\%$ in the initial phase 2 validation. However, this needs to be further justified in a larger training and validation cohort.

Comment 1c. Was there reproducibility of the assays?

Response 1c: Please refer to response 1a.

Comment 1d. Could the authors also give a sense of the ability to automate the steps outlines in sample preparation?

Response 1d: The purification of HCC EVs using EV Click Chip can be automated by adopting a fluidic handler originally developed for our NanoVelcro CTC Assay. After incubating plasma samples with the TCO conjugated antibody cocktail, the plasma samples can be introduced into the EV Click Chips, where the HCC EV purification can be conducted in an automated fashion. The fluidic handler houses a chip holder that allow for convenient assembly of the Tz-grafted SiNWS and PDMS chaotic mixer into an EV Click Chip. There are alignment markers on the lower piece to

allow instant assembly of the devices. Further, the fluidic handler is designed to hold an EV Click Chip together and simultaneously connect with a syringe/syringe pump (for controlling the delivery and flow of plasma and reagents at a designated flow rate). It takes <1 min to assemble the device prior to conducting HCC EV purification experiments. Moreover, we used BioRad RT-ddPCR for the downstream molecular analysis. QX200 AutoDG Droplet Digital PCR System is commercially available automated solution for performing the RT-ddPCR assay.

Comment 1e. Could this be feasibly rolled out in a clinical setting?

Response 1e: Given the rapidity of the processing and ddPCR analysis, the resulting assay is feasible as a clinical test. We appreciate the pathway to development of a clinical-grade test is long and will require controlled validation. But based on our initial experience we believe that our findings merit such attention. We are planning a more comprehensive study using training and validation cohorts for detecting early-stage HCC from at-risk cirrhotic patients to confirm and validate our assays.

Comment 2: The authors list no limitations to this methodology in the discussion? Issues around reproducibility and ensuring that these findings are applicable across disease etiologies are important, especially since they are targeting a panel of tumor specific genes that may not be present in all etiologies of HCC. Future validation studies, including larger Phase 2 biomarker validation and Phase 3 biomarker validation should be discussed.

Response 2: Thanks for the reviewer's suggestion. We have listed the limitations in the discussion, and future plans for larger Phase 2 biomarker validation and Phase 3 "Retrospective Longitudinal" biomarker validation are discussed. We have also added the experimental results focusing on the reproducibility and initially demonstrating that this assay is applicable across all etiologies of HCC.

Comment 3: Please provide tumor size units for Supplemental Table 1. Also, were control samples with cirrhosis confirmed not to have HCC? Please provide details on how these patients were adjudicated as controls.

Response 3: We appreciate this suggestion and have added the tumor size units (cm) in **Supplemental Table 1**. We confirmed that cirrhosis control did not have HCC at the time of blood draw based on 1) negative multiphasic CT/MRI results, or 2) negative liver ultrasound result at the time of blood draw and 6 months follow up or 3) no evidence of HCC on liver explant. These details are provided in the manuscript, **Lines 274-277, Page 15**.

Comment 4: Please provide additional details for how the clinical samples were stored. Was there only one freeze-thaw cycle?

Response 4: The plasma samples are aliquoted and stored in -80°C refrigerators (added in the method section in manuscript **Lines 552 and 555-556, Page 28**). All plasma samples subjected to EV Click Chip and downstream RT-ddPCR assay underwent only one freeze-thaw cycle.

Comment 5: There is no racial information on the participants. Please provide this to get a sense in the heterogeneity of the samples

Response 5: We thank the reviewer for this suggestion and have added the racial information of the participants in the **Supplementary Table S4-S7**.

REVIEWER COMMENTS

Reviewer #1 (Remarks to the Author):

The authors have adequately addressed most of the comments. They performed a substantial amount of work which resulted in a much improved revised manuscript. Nevertheless I still have two remaining comments related to the answers formulated by the authors.

1) Answer to comment 1: The new data do not demonstrate the presence of SRY and C1orf101 transcripts in EVs. The most simple experiment would be that the authors treat HepG2 EVs with first protease followed by RNase and then perform RT-qPCR to demonstrate that SRY and C1orf101 transcripts are enclosed inside EVs. RNase treatment only, as currently performed by the authors, does not degrade RNA that is protected by proteins (as is the case for ribonucleoprotein complexes).

2) Answer to comment 4 (manuscript line 290): The authors indicate multiple databases as evidence that the 10 mRNAs are present in EVs. However it is very important to acknowledge that data enclosed in these databases are obtained from different research groups using different methods that separate EVs with different specificity (PMID:32002166 and PMID:30395310). In other words, it is not because an mRNA is listed in this database that it is truly EV-associated. Please reconsider this phrasing. Furthermore these 10 mRNAs could be easily confirmed on HepG2 EVs using a similar experimental set-up as described above: first protease treatment followed by RNase treatment and RT-qPCR for those mRNAs.

Reviewer #2 (Remarks to the Author):

The authors have done an excellent job of responding to all comments raised by the 3 reviewers. I would recommend publication.

Reviewer #3 (Remarks to the Author):

The authors have addressed most of my comments. The clinical reliability assessment only consisted of 5 samples and this should be noted as a limitation of the study as well in the discussion.

Response to reviewer

Reviewer #1 (Remarks to the Author):

The authors have adequately addressed most of the comments. They performed a substantial amount of work which resulted in a much improved revised manuscript. Nevertheless I still have two remaining comments related to the answers formulated by the authors.

1) Answer to comment 1: The new data do not demonstrate the presence of SRY and C1orf101 transcripts in EVs. The most simple experiment would be that the authors treat HepG2 EVs with first protease followed by RNase and then perform RT-qPCR to demonstrate that SRY and C1orf101 transcripts are enclosed inside EVs. RNase treatment only, as currently performed by the authors, does not degrade RNA that is protected by proteins (as is the case for ribonucleoprotein complexes).

Response 1: We thank the reviewer for his/her comment regarding “demonstrating the presence of SRY and C1orf101 transcripts in EVs”. Per the reviewer’s suggestion, we carried out the experiment depicted in the workflow shown below. In short, HepG2 EVs were first treated with protease to digest ribonucleoprotein. After inactivating protease (at 90°C), RNase was added to the mixture to digest free RNA. The mixture was then subjected to EV lysis and RNA extraction, followed by RT-ddPCR to detect SRY and C1orf101 transcripts along with the 10 HCC-specific genes. In parallel, we carried out the same study using Hep3B EVs (obtained from an SRY-negative HCC cell line). Here (circled in Red), we are delighted to report that SRY and C1orf101 transcripts were detected in HepG2 EVs after treatments by protease and RNase, and only C1orf101 transcript was detected in Hep3B EVs. We have summarized the aforementioned experimental procedure and results in **supporting information 8 (Lines 229-235, Page 15)** and **Supplementary Figure S8b**.

Supplementary Figure S8b. Signals of SRY/Corf101 transcripts and the 10 genes in HepG2-derived EVs and Hep3B-derived EVs, both treated with protease followed by RNase.

2) Answer to comment 4 (manuscript line 290): The authors indicate multiple databases as evidence that the 10 mRNAs are present in EVs. However it is very important to acknowledge that data enclosed in these databases are obtained from different research groups using different methods that separate EVs with different specificity (PMID:32002166 and PMID:30395310). In other words, it is not because an mRNA is listed in this database that it is truly EV-associated. Please reconsider this phrasing. Furthermore these 10 mRNAs could be easily confirmed on HepG2 EVs using a similar experimental set-up as described above: first protease treatment followed by RNase treatment and RT-qPCR for those mRNAs.

Response 2: We appreciate the reviewer’s comments. According to the reviewer’s suggestions, we have revised the description of the expression levels of the 10 mRNA in EV databases in the manuscript (Lines 289-291, Page 15). Furthermore, the experiments summarized in **Response 1**

confirmed that the 10 mRNAs (Circled in Green) can be detected in both HepG2 EVs and Hep3B EVs following the reviewer's suggestion (to treat both HepG2 EVs and Hep3B EVs with protease and RNase before RT-ddPCR). Again, the experimental procedure and results were summarized in supporting information 8 (Lines 229-235, Page 15) and Supplementary Figure S8b.

Reviewer #3 (Remarks to the Author)

The authors have addressed most of my comments. The clinical reliability assessment only consisted of 5 samples and this should be noted as a limitation of the study as well in the discussion.

Response: Thanks for the reviewer's suggestion. We have added this limitation to the discussion in manuscript (Lines 426-427, Page 22).

REVIEWERS' COMMENTS:

Reviewer #1 (Remarks to the Author):

I want to congratulate the authors for the effort to perform the remaining experiments, which further strengthens the manuscript. The authors have appropriately addressed all my concerns. I have no further comments.

RESPONSE TO REVIEWERS' COMMENTS:

Reviewer #1 (Remarks to the Author):

I want to congratulate the authors for the effort to perform the remaining experiments, which further strenghtens the manuscript. The authors have appropriately addressed all my concerns. I have no further comments.

Thank you very much.